# VARIATIONAL SEMANTIC DECOMPOSITION OF COMPOSITIONAL REPRESENTATIONS

## ABSTRACT

Humans exploit the world's compositional structure by combining familiar concepts to form new ones. A key goal in artificial intelligence is to model this capability by learning compositional representations that reflect the structure of the world, and its inverse operation, semantic decomposition, which involves recovering meaningful constituents from composite representations to enable generalization. The binding operation provides a way to compose vector representations, but existing decomposition methods, such as resonator networks, are limited by non-differentiable dynamics, have no guarantees for convergence, and assume quasi-orthogonality of constituent factors. We introduce the Variational Resonator Network (VRN), which reframes decomposition as Bayesian inference. The VRN is a fully differentiable model derived from a variational free energy objective, which allows it to be integrated into neural architectures. It generalizes decomposition beyond representations generated from quasi-orthogonal bipolar vectors and is guaranteed to converge. Experiments show that VRN is comparable to the resonator network on decomposition accuracy when factors are quasi-orthogonal and outperforms it when factors are correlated or real, while also integrating naturally into probabilistic models.

## 1 INTRODUCTION

The world as we know it exhibits compositional structure. Humans exploit this structure when navigating the world and understanding new situations by composing old concepts to form new ones. For example, we understand "red convertible" by composing the concepts of "red" and "convertible." This ability to build complex meanings from a vocabulary of simpler parts is central to how we reason, communicate, and generalize our knowledge to new situations.

Similarly, a fundamental goal in artificial intelligence is to model this capability by learning compositional representations that reflect the structure of the world. Compositionality has been explored through various directions, including Bayesian program learning (Ellis et al., 2020), neuro-symbolic approaches (Tsamoura et al., 2021; Sehgal et al., 2024; Hersche et al., 2023), composable diffusion models (Liu et al., 2023; Du et al., 2024), and object-centric learning (Locatello et al., 2020). Dual to the process of composition is its inverse, i.e., semantic decomposition. Semantic decomposition involves identifying or learning the meaningful constituent parts of a given composite representation. This process is essential as complex representations need to be decomposed in order to be used; a model that can isolate factors such as an object's color, shape, and position can generalize better to unseen combinations.

Notably, one method to generate composite representations is through the binding operation. Given two vectors, one can generate a conjunctive representation using the binding operator, which is typically implemented as element-wise multiplication. Thus, binding enables vectors to be composed into complex representations, and has been used to construct vector representations for data structures such as sequences, sets, and graphs (Kleyko et al., 2023). In particular, when these vector representations are high-dimensional and distributed, these properties can be exploited to decode these composite representations into their constituents. Namely, Frady et al. (2020) propose resonator networks as a solution to this problem of decomposition.

While resonator networks are effective in solving the decomposition problem for quasi-orthogonal codebooks of bipolar vectors, i.e., vectors $\mathbf{x} \in \{-1, 1\}^D$, they have highly non-linear dynamics

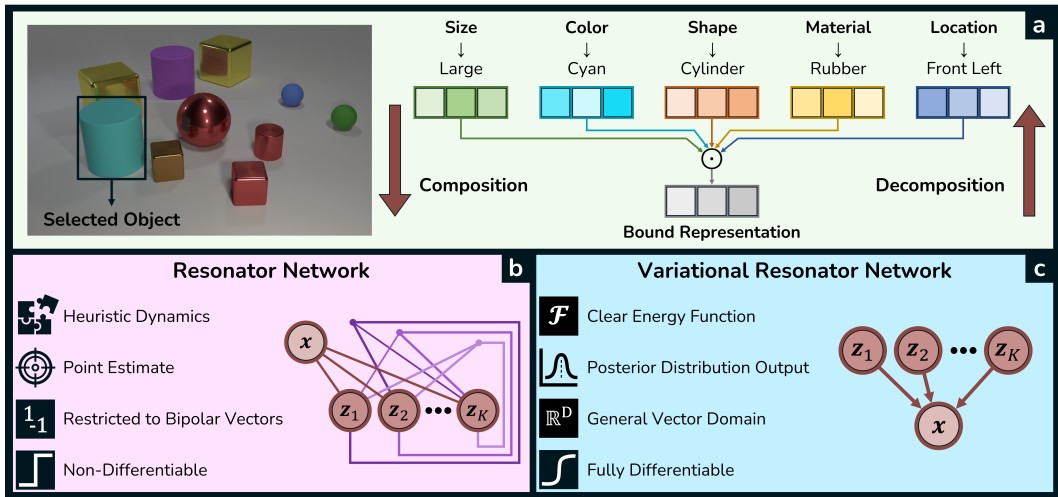

Figure 1: **(a)** Binding-based composition/decomposition: feature vectors (size, color, shape, material, location) are bound to form a single representation; decomposing this representation involves an exponential search. **(b)** Resonator networks were proposed as a solution to the decomposition problem but, in their original form, operate only on bipolar vectors, are non-differentiable, and lack a clear energy function. **(c)** Our Variational Resonator Network (VRN) defines an explicit energy function and performs variational inference, works over general vector spaces, and is fully differentiable, enabling integration with neural architectures.

and are not guaranteed to converge. This limits their applicability in deep learning applications that require clear gradient information for backpropagation. In addition, resonator networks require the codebook vectors to be quasi-orthogonal, but there exist applications in which vector representations are correlated. To overcome these limitations, we propose a new framework that reframes semantic decomposition as Bayesian inference. Our approach, the Variational Resonator Network (VRN), infers a posterior distribution over the latent factors that could have generated a bound vector representation. The VRN is a fully differentiable model derived from a variational free energy objective, which allows it to be integrated into deep learning systems. It generalizes factorization beyond representations generated from quasi-orthogonal bipolar vectors and the inference process is guaranteed to converge. Figure 1 visualizes composition and decomposition via binding, and summarizes the characteristics of the resonator network and VRN.

Our contributions are as follows: (1) We reframe the decomposition problem as Bayesian inference, and propose the Variational Resonator Network as a solution to the problem. (2) We show comparable performance to resonator networks on decomposition with quasi-orthogonal codebooks and superior performance on decomposition with correlated codebooks. Notably, resonator networks have not been evaluated in the correlated regime to the best of our knowledge. (3) We demonstrate how our method can be integrated into existing probabilistic models.

## 2 BACKGROUND

### 2.1 RESONATOR NETWORKS

**The Decomposition Problem**   The decomposition problem is the problem of recovering the constituent vectors of a composite vector generated through the binding operation. Suppose we have $K$ codebooks, each with $n$ bipolar vectors of dimension $D$, i.e., $\mathbf{X}_1, \dots, \mathbf{X}_K \in \{-1, 1\}^{D \times n}$. Moreover, assume that the vectors are quasi-orthogonal in each codebook $\mathbf{X}_j$, i.e., $\frac{\mathbf{x}^\top \mathbf{y}}{D} \approx 0$ for $\mathbf{x}, \mathbf{y} \in \mathbf{X}_j$. (We overload $\in$ for convenience.) Let $\mathbf{x}_j \in \mathbf{X}_j$ be some choice of vector from the $j$-th codebook, for $j = 1, \dots, K$. Let $\mathbf{x} = \mathbf{x}_1 \odot \cdots \odot \mathbf{x}_K \in \{-1, 1\}^D$ be the composite vector obtained through binding (i.e. element-wise multiplication). Given the composite vector $\mathbf{x}$ and codebooks $\mathbf{X}_1, \dots, \mathbf{X}_K$, we would like to recover the constituent vectors $\mathbf{x}_1, \dots, \mathbf{x}_K$. This process involves a

combinatorial search problem whose search space is exponential in the number of factors, i.e., $n^K$. The problem can be generalized beyond bipolar vectors, e.g., vectors in $\mathbb{R}^D$.

**Resonator Networks**    Resonator networks were introduced as an approximate method for solving the decomposition problem. The main idea behind the resonator network is to recurrently apply an update rule on estimates $\hat{\mathbf{x}}_1, \ldots, \hat{\mathbf{x}}_K$:

$$\hat{\mathbf{x}}_j^{t+1} = \mathrm{sgn}(\mathbf{X}_j \mathbf{X}_j^\top (\mathbf{x} \odot \mathbf{r}_j^t)), \quad \text{for } j = 1, \ldots, K \tag{1}$$

where $\mathbf{r}_j^t = \bigodot_{i \neq j} \mathbf{x}_i^t$ and $\mathbf{x}_j^0 = \frac{1}{n} \sum_{\mathbf{x}_j \in \mathbf{X}_j} \mathbf{x}_j$ for $j = 1, \ldots, K$.

Notice that this update rule is reminiscent of the update rule in Hopfield networks (Hopfield, 1982):

$$\hat{\mathbf{x}}_{t+1} = \mathrm{sgn}(\mathbf{X}\mathbf{X}^\top \hat{\mathbf{x}}_t) \tag{2}$$

Hopfield networks are auto-associative memories where the vectors in the codebook $\mathbf{X}$ form fixed-point attractors in the dynamics defined by the update rule, provided that the number of memorized vectors is below the capacity of network. If each codebook vector is a random pattern where each element is sampled uniformly from $\{-1, 1\}$, it has been shown that the memory capacity of network is approximately $0.138D$, where $D$ is the dimension of the vector (Amit et al., 1985). Modern variants have improved capacity exponential in $D$ (Krotov & Hopfield, 2016; Ramsauer et al., 2021).

Thus, resonator networks can be seen as $K$ coupled Hopfield networks, where the $j$-th network has the codebook vectors $\mathbf{x} \in \mathbf{X}_j$ as fixed-point attractors. Under the assumption each Hopfield network is not over capacity, i.e. each $\mathbf{x} \in \mathbf{X}_j$ is an attractor in the $j$-th network for $j = 1, \ldots, K$, the correct solution to the decomposition problem $\hat{\mathbf{x}}_j = \mathbf{x}_j$ for $j = 1, \ldots, K$ is a fixed point in the dynamical system defined by the resonator network update rule, i.e.

$$\mathbf{x}_j = \mathrm{sgn}\left(\mathbf{X}_j \mathbf{X}_j^\top \left(\mathbf{x} \odot \bigodot_{i \neq j} \mathbf{x}_i\right)\right), \quad \text{for } j = 1, \ldots, K \tag{3}$$

This result follows from the fact that $\mathbf{x} \odot \bigodot_{i \neq j} \mathbf{x}_i = \mathbf{x}_j$ as $\mathbf{x} = \bigodot_{i=1}^{K} \mathbf{x}_i$ and each $\mathbf{x}_i \in \{-1, 1\}^D$ is its own multiplicative inverse. From this, we can see that the purpose of $\mathbf{x} \odot \mathbf{r}_j^t$ in the resonator update rule (Eq. 1) is to estimate the $j$-th factor *given* the remaining factors.

While Hopfield networks minimize a known energy function, it is not known what energy function resonator networks minimize. Moreover, the dynamics of resonator networks occasionally enters limit cycles and does not converge.

## 3   VARIATIONAL RESONATOR NETWORK

**Decomposition as Bayesian Inference**    We frame the decomposition problem from the perspective of Bayesian inference. For now, let us consider vectors in $\{-1, 1\}^D$ and assume $K = 2$. Let $\mathbf{x} \in \{-1, 1\}^D$ be a composite vector obtained by binding two factors. Let $\mathbf{z}_1, \mathbf{z}_2 \in \mathbb{R}^n$ be one-hot vectors with respective codebooks $\mathbf{X}_1, \mathbf{X}_2 \in \{-1, 1\}^{D \times n}$. Assume each entry of the codebooks is sampled independently from $\mathrm{Unif}(\{-1, 1\})$. We can model $\mathbf{x}$, $\mathbf{z}_1$, and $\mathbf{z}_2$ as random variables. In particular, suppose we have the generative model (Figure 2a)

$$p(\mathbf{x}, \mathbf{z}_1, \mathbf{z}_2) = p(\mathbf{x}|\mathbf{z}_1, \mathbf{z}_2)p(\mathbf{z}_1)p(\mathbf{z}_2) \tag{4}$$

We can frame the decomposition problem as inferring $\mathbf{z}_1$ and $\mathbf{z}_2$ by finding the posterior $p(\mathbf{z}_1, \mathbf{z}_2|\mathbf{x})$. We do this by introducing a variational posterior $q_\phi(\mathbf{z}_1, \mathbf{z}_2|\mathbf{x})$ and performing variational inference. We can use a mean-field approximation for the variational posterior (Figure 2b)

$$q_\phi(\mathbf{z}_1, \mathbf{z}_2|\mathbf{x}) = q_{\phi_1}(\mathbf{z}_1|\mathbf{x})q_{\phi_2}(\mathbf{z}_2|\mathbf{x}) \tag{5}$$

Since $\mathbf{z}_1$ and $\mathbf{z}_2$ specify choices of codebook vectors in codebooks $\mathbf{X}_1$ and $\mathbf{X}_2$ respectively, we let $q_{\phi_1}(\mathbf{z}_1|\mathbf{x})$ and $q_{\phi_2}(\mathbf{z}_2|\mathbf{x})$ be categorical distributions with logits $\phi_1$ and $\phi_2$ respectively.

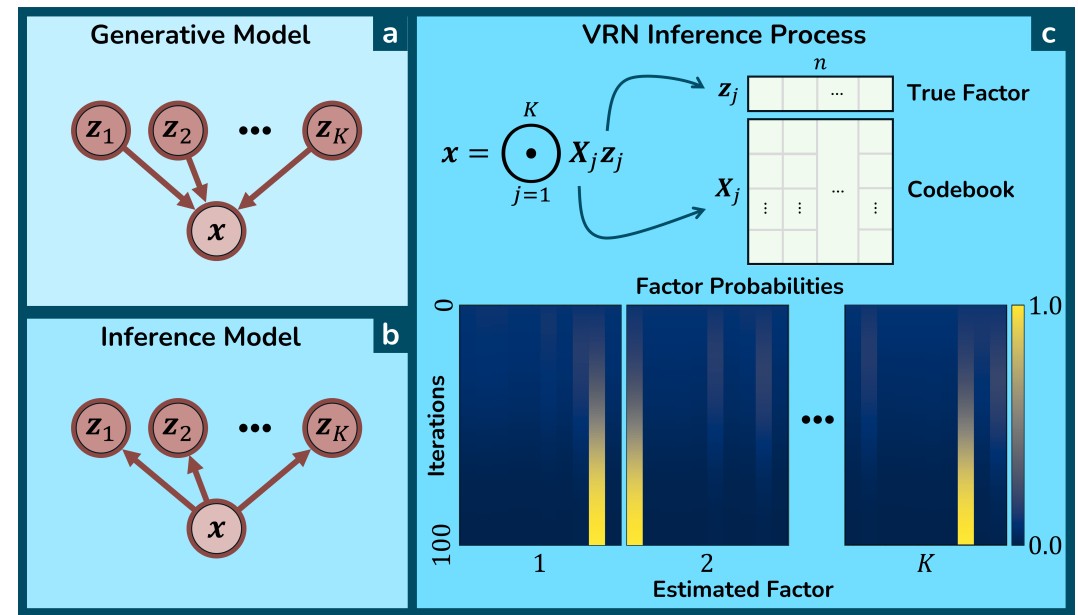

Figure 2: Graphical model for the **(a)** generative and **(b)** inference processes of the VRN. **(c)** Top: We generate a composite representation via the binding operator. Bottom: a visualization of the estimated factors converging to the true factors as variational free energy is optimized.

**Loss Function**    With this approach, to infer $\mathbf{z}_1$ and $\mathbf{z}_2$, we want to minimize the variational free energy (equivalently, the negative ELBO)

$$\mathcal{L}(\phi) = \mathbb{E}_{q_\phi(\mathbf{z}_1, \mathbf{z}_2|\mathbf{x})}[\log q_\phi(\mathbf{z}_1, \mathbf{z}_2|\mathbf{x}) - \log p(\mathbf{x}, \mathbf{z}_1, \mathbf{z}_2)] \tag{6}$$

Suppose the likelihood is given by

$$p(\mathbf{x}|\mathbf{z}_1, \mathbf{z}_2) = \frac{\exp \delta(\mathbf{x}, \mathbf{X}_1\mathbf{z}_1 \odot \mathbf{X}_2\mathbf{z}_2)}{\sum_{\mathbf{z}_1'} \sum_{\mathbf{z}_2'} \exp \delta(\mathbf{X}_1\mathbf{z}_1' \odot \mathbf{X}_2\mathbf{z}_2', \mathbf{X}_1\mathbf{z}_1 \odot \mathbf{X}_2\mathbf{z}_2)} \tag{7}$$

where the normalization is over choices of codebook vectors $\mathbf{z}_1'$ and $\mathbf{z}_2'$. $\delta$ is a similarity function defined by $\delta(\mathbf{x}, \mathbf{y}) = \mathbf{x}^\top \mathbf{y}/D$. Let $\mathbf{p}_1 = \mathrm{softmax}(\phi_1)$ and $\mathbf{p}_2 = \mathrm{softmax}(\phi_2)$. Then

$$\log q_\phi(\mathbf{z}_1, \mathbf{z}_2|\mathbf{x}) = \log(\mathbf{p}_1^\top \mathbf{z}_1) + \log(\mathbf{p}_2^\top \mathbf{z}_2) \tag{8}$$

$$\log p(\mathbf{x}, \mathbf{z}_1, \mathbf{z}_2) = \log p(\mathbf{x}|\mathbf{z}_1, \mathbf{z}_2) + \log p(\mathbf{z}_1) + \log p(\mathbf{z}_2) \tag{9}$$

$$= \delta(\mathbf{x}, \mathbf{X}_1\mathbf{z}_1 \odot \mathbf{X}_2\mathbf{z}_2) \tag{10}$$

$$- \log \sum_{\mathbf{z}_1', \mathbf{z}_2'} \exp\left[\delta(\mathbf{X}_1\mathbf{z}_1' \odot \mathbf{X}_2\mathbf{z}_2', \mathbf{X}_1\mathbf{z}_1 \odot \mathbf{X}_2\mathbf{z}_2)\right] \tag{11}$$

$$+ \log p(\mathbf{z}_1) + \log p(\mathbf{z}_2) \tag{12}$$

Expression 11 involves a sum exponential in $K$, making it generally intractable. We make the following proposition which allows us to simplify the loss:

**Proposition 1** (Quasi-orthogonality under binding). *Let $\{\mathbf{X}_j \in \{-1, 1\}^{D \times n}\}_{j=1}^K$ with independent entries $[\mathbf{X}_j]_{kl} \sim \mathrm{Unif}(\{-1, 1\})$ for $k = 1, \ldots, D$ and $l = 1, \ldots, n$. Pick columns $\mathbf{x}_j, \mathbf{y}_j \in \mathbf{X}_j$ for $j = 1, \ldots, K$. Define $\delta(\mathbf{x}, \mathbf{y}) = \mathbf{x}^\top \mathbf{y}/D$. Then*

$$\delta\left(\bigodot_{j=1}^K \mathbf{x}_j, \bigodot_{j=1}^K \mathbf{y}_j\right) = 1 \quad \textit{if } \mathbf{x}_j = \mathbf{y}_j \textit{ for } j = 1, \ldots, K, \tag{13}$$

$$\delta\left(\bigodot_{j=1}^K \mathbf{x}_j, \bigodot_{j=1}^K \mathbf{y}_j\right) \to 0 \textit{ as } D \to \infty \quad \textit{otherwise.} \tag{14}$$

*Proof.* See Appendix A. □

As a result of Proposition 1, we have the following corollary:

**Corollary 2.** $\sum_{\mathbf{z}_1'} \sum_{\mathbf{z}_2'} \exp \delta(\mathbf{X}_1 \mathbf{z}_1' \odot \mathbf{X}_2 \mathbf{z}_2', \mathbf{X}_1 \mathbf{z}_1 \odot \mathbf{X}_2 \mathbf{z}_2)$ *is asymptotically constant as* $D \to \infty$.

Thus, Corollary 2 enables us to ignore expression 11 for sufficiently large $D$. In practice, we take $D$ to be relatively large (e.g., $D = 1000$). This results in the loss

$$\mathcal{L}(\phi) = \mathbb{E}_{q_\phi}[\log(\mathbf{p}_1^\top \mathbf{z}_1) + \log(\mathbf{p}_2^\top \mathbf{z}_2) - \delta(\mathbf{x}, \mathbf{X}_1 \mathbf{z}_1 \odot \mathbf{X}_2 \mathbf{z}_2) - \log p(\mathbf{z}_1) - \log p(\mathbf{z}_2)] \quad (15)$$

$$= -\delta(\mathbf{x}, \mathbf{X}_1 \mathbf{p}_1 \odot \mathbf{X}_2 \mathbf{p}_2) + \sum_{j=1}^{2} D_{\mathrm{KL}}[q_{\phi_j}(\mathbf{z}_j|\mathbf{x}) \| p(\mathbf{z}_j)] \quad (16)$$

The simplification of the expectation is due to the bilinearity of $\delta$. If we further assume $p(\mathbf{z}_1)$ and $p(\mathbf{z}_2)$ are uniform, we have the loss

$$\mathcal{L}(\phi) = -\delta(\mathbf{x}, \mathbf{X}_1 \mathbf{p}_1 \odot \mathbf{X}_2 \mathbf{p}_2) - H(\mathbf{p}_1) - H(\mathbf{p}_2) \quad (17)$$

The result of this optimization process is visualized in Figure 2c. We can illustrate the similarity to the resonator update rule in Eq. 1 by taking the gradient with respect to $\mathbf{p}_1$ and $\mathbf{p}_2$:

$$\nabla_{\mathbf{p}_1} \mathcal{L}(\phi) = -\frac{1}{D} \mathbf{X}_1^\top (\mathbf{x} \odot \mathbf{X}_2 \mathbf{p}_2) - \log \mathbf{p}_1 - \mathbf{1}$$
$$\nabla_{\mathbf{p}_2} \mathcal{L}(\phi) = -\frac{1}{D} \mathbf{X}_2^\top (\mathbf{x} \odot \mathbf{X}_1 \mathbf{p}_1) - \log \mathbf{p}_2 - \mathbf{1} \quad (18)$$

We include a more detailed derivation of this result in Appendix C.

**Generalization to $K$ Factors** Suppose we instead have $K$ codebooks $\mathbf{X}_1, \ldots, \mathbf{X}_K \in \{-1, 1\}^{D \times n}$ with corresponding one-hot vectors $\mathbf{z}_1, \ldots, \mathbf{z}_K \in \mathbb{R}^n$. Let $\mathbf{x} \in \{-1, 1\}^D$ be a composite vector formed by binding one vector from each of the $K$ codebooks. Following the same process, the loss function is

$$\mathcal{L}(\phi) = -\delta \left( \mathbf{x}, \bigodot_{j=1}^{K} \mathbf{X}_j \mathbf{p}_j \right) + \sum_{j=1}^{K} D_{\mathrm{KL}}[q_{\phi_j}(\mathbf{z}_j|\mathbf{x}) \| p(\mathbf{z}_j)] \quad (19)$$

where $\mathbf{p}_j = \mathrm{softmax}(\phi_j), j = 1, \ldots, K$.

**Extension to Real Factors** An analogous result to Proposition 1 holds when codebooks have entries sampled from $\mathcal{N}(0, 1)$, which extends quasi-orthogonality to vectors in $\mathbb{R}^D$. (See Proposition 4 in Appendix A.)

**Computational Complexity** The computational cost is dominated by the gradient calculation of the similarity reconstruction term. This operation requires computing matrix-vector products between the codebooks $\mathbf{X}_j$ of size $D \times n$ and the current probability estimates of the other factors. For $K$ factors, this results in a per-iteration complexity of $O(DKn)$. Crucially, this is identical to the cost of the standard resonator network update rule, implying that our variational framework introduces no additional computational overhead.

## 4 RESULTS

### 4.1 DECOMPOSITION OF QUASI-ORTHOGONAL VECTORS

We compare the decomposition accuracy of resonator networks against the variational resonator network. We consider the decomposition problem with quasi-orthogonal codebooks; i.e., we randomly generate codebooks $\mathbf{X}_1, \ldots, \mathbf{X}_K \in \{-1, 1\}^{D \times n}$ of size $n$ such that each entry is sampled i.i.d. from $\mathrm{Unif}(\{-1, 1\})$. We choose some arbitrary bound hypervector $\mathbf{x} = \mathbf{x}_1 \odot \cdots \odot \mathbf{x}_K$, where $\mathbf{x}_j$ is chosen uniformly randomly from the columns of $\mathbf{X}_j$. We compute the accuracy of a single composite

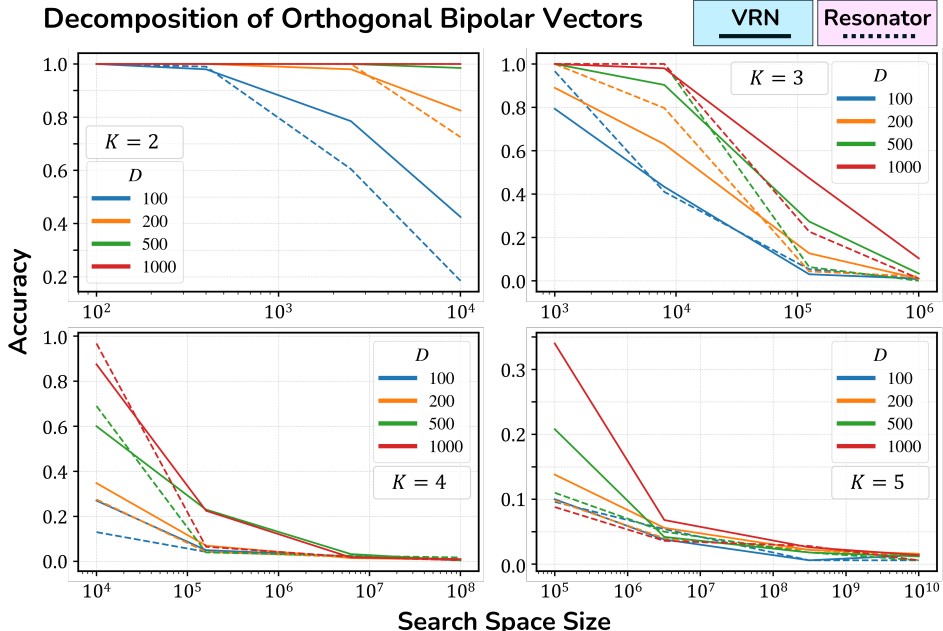

Figure 3: Decomposition accuracy of resonator network (dotted line) and VRN (solid line) over varying search space sizes $n^K$. Here, we consider decomposition of quasi-orthogonal bipolar vectors. Different line colors correspond to different dimensions $D$. Each plot corresponds to a different number of codebooks ($K = 2, 3, 4, 5$).

vector as $c/K$ where $c$ is the number of correct factors. We take the average over 100 samples to compute the accuracy.

Figure 3 visualizes the decomposition accuracy as a function of the search space $n^K$ for different number of factors $K$. The VRN achieves comparable performance to resonator networks, but degrades in accuracy more gracefully as the search space increases. (See Appendix F for experimental details.)

## 4.2 DECOMPOSITION OF CORRELATED VECTORS

While the decomposition problem was originally formulated for quasi-orthogonal codebooks, there are many cases in which codebooks contain correlated representations (e.g., words with similar meanings should have representations that are more similar than those with disparate meanings.) Thus, we break the assumption of quasi-orthogonality and examine the performance of the resonator network and VRN on correlated codebooks.

To generate correlated bipolar codebooks with correlation $\rho$, we start off with a single randomly sampled bipolar vector $\mathbf{x}$. For each of the $n$ vectors in the codebook, we take $\mathbf{x}$ and flip each entry by probability $p = (1 - \rho)/2$. This results in codebook vectors with pairwise correlation $\rho^2$.

Figure 4 visualizes the decomposition accuracy over different levels of codebook correlation for different numbers of factors $K$. The VRN outperforms the resonator network in all cases, while both models benefit from increased correlation.

## 4.3 REAL VS BIPOLAR CODEBOOKS

We consider codebooks with vectors in $\mathbb{R}^D$. We extend the resonator network to this case by replacing the sgn activation with tanh in Eq. 1. (See Appendix G for other choices of activations.) Figure 5 shows that VRN consistently outperforms resonator network in this case. We also compare the performance between real and bipolar codebooks in Figure 6. Bipolar representations generally

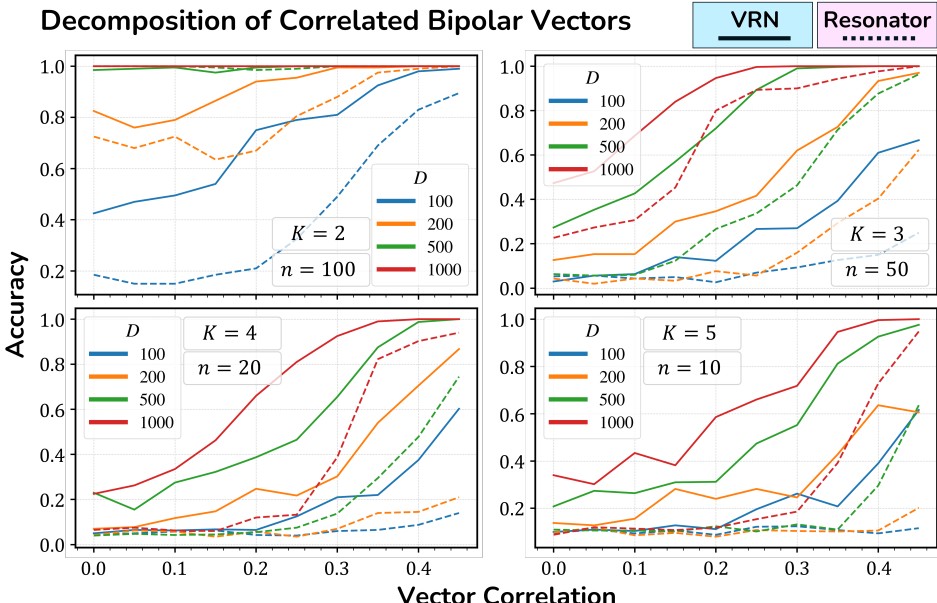

Figure 4: Decomposition accuracy of resonator network (dotted line) and VRN (solid line) over levels of correlation. Different line colors correspond to different dimensions $D$. Each plot corresponds to a different number of codebooks ($K = 2, 3, 4, 5$).

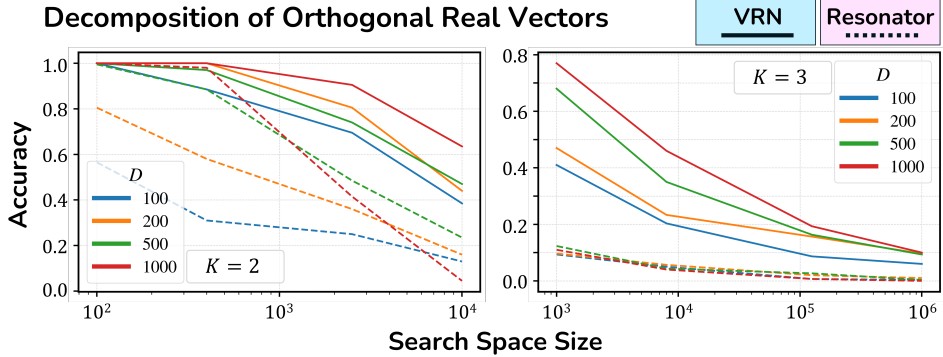

Figure 5: Decomposition accuracy of resonator network (dotted line) and VRN (solid line) over varying search space sizes $n^K$. Here, we consider decomposition of quasi-orthogonal real vectors. Different line colors correspond to different dimensions $D$. Each plot corresponds to a different number of codebooks ($K = 2, 3$).

perform better for smaller search spaces where the accuracy is high, but drop to similarly low levels as the model saturates. Similar to bipolar vectors, VRN significantly outperforms resonator network for correlated real codebooks (see Figure 9 in Appendix E).

## 4.4 VARIATIONAL RESONATOR AUTOENCODER

One benefit of framing semantic decomposition as variational inference is its easy integration into probabilistic models. As an example, we can expand the framework to generate data $\mathbf{y} \sim p(\mathbf{y})$. Let $\mathbf{z} = [\mathbf{z}_1, \ldots, \mathbf{z}_K]$, with $\mathbf{z}_j$ as above. Consider the generative model and variational posterior

$$p_\theta(\mathbf{y}, \mathbf{x}, \mathbf{z}) = p_\theta(\mathbf{y}|\mathbf{x})p(\mathbf{x}|\mathbf{z})p(\mathbf{z}), \quad q_\phi(\mathbf{x}, \mathbf{z}|\mathbf{y}) = q_\phi(\mathbf{z}|\mathbf{x})q_\psi(\mathbf{x}|\mathbf{y}) \tag{20}$$

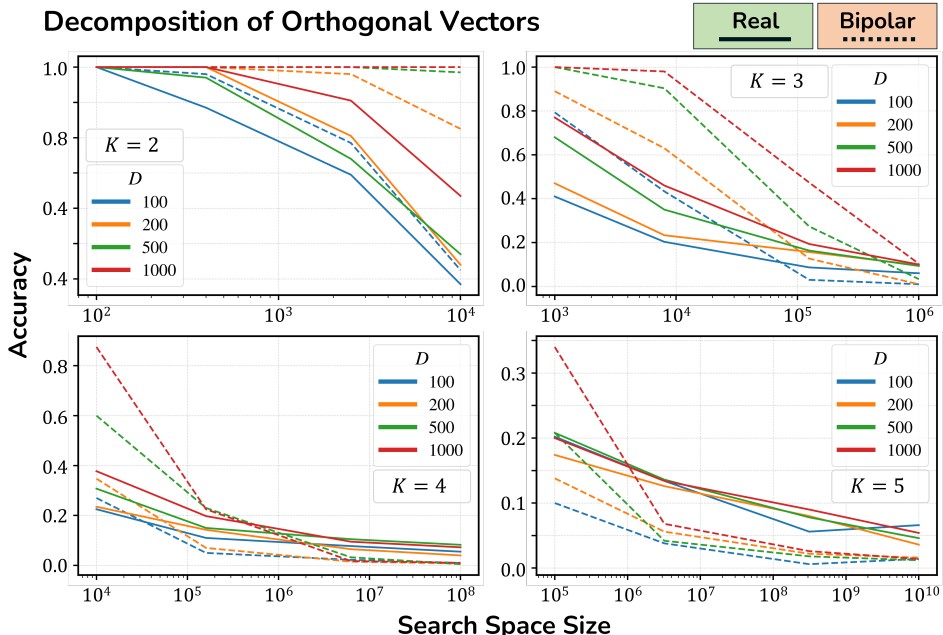

Figure 6: Decomposition accuracy of bipolar (dotted line) and real (solid line) codebooks using VRN over varying search space sizes $n^K$. Different line colors correspond to different dimensions $D$. Each plot corresponds to a different number of codebooks ($K = 2, 3, 4, 5$).

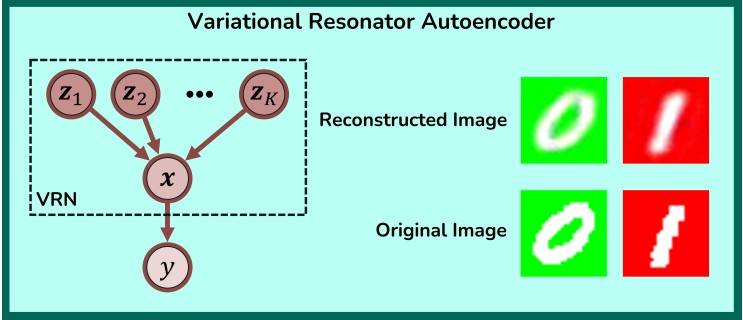

Figure 7: **Left**: Graphical model for the Variational Resonator Autoencoder (VRAE), which includes the VRN as a subcomponent. **Right**: colorized-MNIST images reconstructed by the VRAE.

Specifically, $p(\mathbf{x}|\mathbf{z})$ and $q_\phi(\mathbf{z}|\mathbf{x})$ are the same as above and $p(\mathbf{z})$ is uniform for each $\mathbf{z}_1, \ldots, \mathbf{z}_K$. For simplicity, we assume that $q_\psi(\mathbf{x}|\mathbf{y}) = \delta_{\text{Dirac}}(\mathbf{x} - f_\psi^{\text{enc}}(\mathbf{y}))$. Then the corresponding loss is

$$\mathcal{L}(\theta, \psi, \phi) = -\log p_\theta(\mathbf{y}|\mathbf{x}) + \mathbb{E}_{q_\phi(\mathbf{z}|\mathbf{x})}[\log q_\phi(\mathbf{z}|\mathbf{x}) - \log p(\mathbf{x}|\mathbf{z})]. \quad (21)$$

We have a deterministic encoder $\mathbf{x} = f_\psi^{\text{enc}}(\mathbf{y})$ as we model the variational posterior with a Dirac-delta. Recall that $q_\phi(\mathbf{z}|\mathbf{x}) = \prod_{j=1}^K q_{\phi_j}(\mathbf{z}_j|\mathbf{x})$. We can amortize the computation of the variational parameters by introducing a parameterized function $h_\lambda(\mathbf{x})$ such that $\phi_j = h_\lambda(\mathbf{x})_j$. This gives us a variant of the variational autoencoder (VAE) (Kingma & Welling, 2014). (See Appendix D for more details.)

Figure 7 shows the generative model for the Variational Resonator Autoencoder (VRAE) as an extension of the VRN. We perform unsupervised training on the VRAE on colorized-MNIST and visualize images and their reconstructions.

## 5 DISCUSSION

**Choice of likelihood**   In our proposed model, we choose a likelihood with a specific parameterization $p(\mathbf{x}|\mathbf{z}_1, \ldots, \mathbf{z}_K) \propto \exp \delta(\mathbf{X}_1\mathbf{z}_1 \odot \cdots \odot \mathbf{X}_K\mathbf{z}_K)$ (Eq. 7) and circumvented normalization through Propositions 1 and 4. The motivating choice behind this likelihood is the bilinearity of $\delta$, which we exploit in the computation of the expectation, allowing us to avoid Monte Carlo estimates and enable direct computation.

A more natural choice would be a Gaussian likelihood, i.e. $p(\mathbf{x}|\mathbf{z}_1, \ldots, \mathbf{z}_K) = \mathcal{N}(\mathbf{X}_1\mathbf{z}_1 \odot \cdots \odot \mathbf{X}_K\mathbf{z}_K, \sigma^2 I)$. However, this choice requires one to use techniques such as the straight-through gradient estimator (Bengio et al., 2013) or the Gumbel-softmax reparameterization trick (Jang et al., 2017) to propagate the gradient through the expectation. In addition, using a Gaussian likelihood did not yield good results empirically. (See Appendix B for more details.)

**Correlated codebooks**   As the decomposition problem was originally formulated for quasi-orthogonal codebooks, the effect of correlation has remained unexplored empirically. Theoretically, correlated vectors contribute to cross-talk noise in the operation of Hopfield networks, increasing the number of spurious attractors and leading to decreased capacity. However, resonator networks see no change or even improved performance as correlation increases.

VRN sees even more significant gains in performance with correlated codebooks. We hypothesize that correlated codebooks make the energy landscape more amenable to optimization. Consider the similarity $\delta(\mathbf{X}_1\mathbf{z}_1 \odot \cdots \odot \mathbf{X}_K\mathbf{z}_K)$. When codebooks are quasi-orthogonal, Proposition 1 suggests this similarity approaches a point-mass function in the limit, making it difficult to optimize. However, one can understand the effect of correlation as "smoothing" this energy landscape, leading to gradients that are more useful for the optimization process.

**Integration into neural architectures**   Representations that are composed via binding commonly fall under the category of Vector Symbolic Architectures (VSAs) (Kleyko et al., 2023). Vector symbolic representations have been integrated into neural architectures for solving practical tasks that involve symbolic manipulation and reasoning. In such scenarios, the VRN can serve as a critical bridge between sub-symbolic neural perception and symbolic reasoning, as a part of a larger pipeline designed for a specific task. For instance, in Neuro-Vector-Symbolic Architectures (NVSA) applied to Raven's Progressive Matrices (Hersche et al., 2023), a neural backbone maps visual panels to composite holographic vectors representing bound attributes (e.g., $\mathbf{x}_{\text{panel}} \approx \mathbf{x}_{\text{shape}} \odot \mathbf{x}_{\text{color}}$). The VRN can function as the decoding module in this pipeline, factorizing these neural embeddings back into their atomic constituents (retrieving the specific "shape" and "color" vectors). These disentangled factors then enable the downstream probabilistic reasoning engine to deduce relationships and solve the matrix. While our current work focuses on optimizing the factorization mechanism itself, this pipeline illustrates the VRN's potential as a robust, differentiable interface for neuro-symbolic systems.

Meanwhile, applications involving resonator networks such as data structure decoding (Frady et al., 2020), visual scene decomposition (Kymn et al., 2024a), and spatial decoding (Kymn et al., 2024b), fix the network parameters due to its highly non-linear dynamics. The VRN offers a solution to

this problem, enabling straightforward integration into neural architectures as shown in Section 4.4. Directions such as the unsupervised learning of disentagled vector symbolic factors, the effect of amortization, and end-to-end encoding and decomposition will be left for future work.

**Representation Learning and Compositionality**   Our work sits at the intersection of probabilistic generative modeling and vector-symbolic compositionality. In the domain of disentangled representation learning, methods such as $\beta$-VAE (Higgins et al., 2016) encourage the emergence of independent factors of variation by heavily regularizing the variational free energy, while VQ-VAE (van den Oord et al., 2017) introduces discrete codebooks to capture categorical latent structures. However, these approaches typically rely on generic non-linear decoders to mix factors, often lacking the explicit, algebraic binding operations required for systematic generalization. To address this, recent neuro-symbolic architectures have introduced explicit structural biases. For instance, the Neural Systematic Binder (Singh et al., 2023) proposes a dedicated binding mechanism to decompose object-centric representations into independent factor slots (e.g., shape, color) via soft- and hard-binding processes. Our VRN shares this goal of explicit semantic decomposition but operates within the holographic domain of VSA.

While $\beta$-VAE and VQ-VAE bear similarities to VRN in their goals, they tackle a fundamentally different problem; they encode input data to latent representation whereas VRN decode structured representations. Additionally, VRN shares the goal of structured decomposition with the Neural Systematic Binder (NSB), but the two approaches diverge fundamentally in their binding mechanisms. NSB employs a concatenative binding strategy, where objects are represented as "block-slots" constructed by spatially routing and concatenating independent factor modules (e.g., separate blocks for shape and color). In contrast, VRN operates on holographic representations (VSA), where factors are algebraically compressed (e.g., via Hadamard products) into a single distributed vector of fixed dimensionality. Consequently, while NSB focuses on the architectural routing of information into slots, VRN solves the inverse problem of disentangling factors that have been superimposed into a shared latent space.

The iterative update dynamics of resonator networks also bear a conceptual resemblance to Adaptive Resonance Theory (ART) (Grossberg, 1976). Similar to how ART relies on "resonance" between bottom-up inputs and top-down expectations to stabilize learning and categorization, the VRN treats decomposition as a dynamic inference process that settles into a stable, energy-minimizing configuration of constituent factors.

## 6 CONCLUSION

We reframed semantic decomposition as Bayesian inference and introduced the Variational Resonator Network (VRN): a fully differentiable, energy-based alternative to resonator networks. By optimizing a variational free energy objective, VRN yields posteriors over latent factors and enables straightforward integration with gradient-based learning. Empirically, VRN achieves comparable performance compared to resonator network under quasi-orthogonal codebooks and clearly outperforms it when codebooks are correlated; it also extends naturally to real-valued codebooks.

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

## A  QUASI-ORTHOGONALITY UNDER BINDING: PROOFS

**Proposition 3** (Quasi-orthogonality under binding). *Let $\{\mathbf{X}_j \in \{-1,1\}^{D \times n}\}_{j=1}^K$ with independent entries $[\mathbf{X}_j]_{kl} \sim \mathrm{Unif}(\{-1,1\})$ for $k = 1, \ldots, D$ and $l = 1, \ldots, n$. Pick columns $\mathbf{x}_j, \mathbf{y}_j \in \mathbf{X}_j$ for $j = 1, \ldots, K$. Define $\delta(\mathbf{x}, \mathbf{y}) = \mathbf{x}^\top \mathbf{y}/D$. Then*

$$\delta\left(\bigodot_{j=1}^K \mathbf{x}_j, \bigodot_{j=1}^K \mathbf{y}_j\right) = 1 \quad \text{if } \mathbf{x}_j = \mathbf{y}_j \text{ for } j = 1, \ldots, K, \tag{22}$$

$$\delta\left(\bigodot_{j=1}^K \mathbf{x}_j, \bigodot_{j=1}^K \mathbf{y}_j\right) \to 0 \text{ as } D \to \infty \quad \text{otherwise.} \tag{23}$$

*Proof.* Let $\mathbf{u} = \mathbf{x}_1 \odot \cdots \odot \mathbf{x}_K$ and $\mathbf{v} = \mathbf{y}_1 \odot \cdots \odot \mathbf{y}_K$. Suppose $\mathbf{x}_j = \mathbf{y}_j$ for $j = 1, \ldots, K$. Then $\mathbf{u} = \mathbf{v} \in \{-1,1\}^D$. It follows that $\delta(\mathbf{u}, \mathbf{v}) = 1$.

If at least one pair differs, then for each coordinate $i$ the product $\prod_{j=1}^K (\mathbf{x}_j)_i (\mathbf{y}_j)_i \sim \mathrm{Unif}(\{-1,1\})$ which has zero mean. So

$$\delta(\mathbf{u}, \mathbf{v}) = \frac{1}{D} \sum_{i=1}^D Z_i$$

for i.i.d. $Z_i \sim \mathrm{Unif}(\{-1,1\})$ with $\mathbb{E}[Z_i] = 0$ and $\mathrm{Var}(Z_i) = 1$. Therefore $\mathbb{E}[\delta(\mathbf{u}, \mathbf{v})] = 0$ and, by the Central Limit Theorem, $\delta(\mathbf{u}, \mathbf{v}) \sim \mathcal{N}(0, 1/\sqrt{D})$ as $D \to \infty$, so $\delta(\mathbf{u}, \mathbf{v}) \to 0$ as $D \to \infty$. $\square$

**Proposition 4** (Quasi-orthogonality under binding: Gaussian Case). *Let $\{\mathbf{X}_j \in \mathbb{R}^{D \times n}\}_{j=1}^K$ with independent entries $[\mathbf{X}_j]_{kl} \sim \mathcal{N}(0,1)$ for $k = 1, \ldots, D$ and $l = 1, \ldots, n$. Pick columns $\mathbf{x}_j, \mathbf{y}_j \in \mathbf{X}_j$ for $j = 1, \ldots, K$. Define $\delta(\mathbf{x}, \mathbf{y}) = \mathbf{x}^\top \mathbf{y}/D$. Then*

$$\delta\left(\bigodot_{j=1}^K \mathbf{x}_j, \bigodot_{j=1}^K \mathbf{y}_j\right) \to \begin{cases} 1 & \text{if } \mathbf{x}_j = \mathbf{y}_j \quad \text{for } j = 1, \ldots, K, \\ 0 & \text{otherwise} \end{cases} \quad \text{as } D \to \infty. \tag{24}$$

*Proof.* Let $\mathbf{u} = \mathbf{x}_1 \odot \cdots \odot \mathbf{x}_K$ and $\mathbf{v} = \mathbf{y}_1 \odot \cdots \odot \mathbf{y}_K$. Suppose $\mathbf{x}_j = \mathbf{y}_j$ for $j = 1, \ldots, K$. Then $\mathbf{u} = \mathbf{v}$. Then as $D \to \infty$,

$$\delta(\mathbf{x}, \mathbf{y}) = \frac{1}{D} \sum_{i=1}^D \mathbf{u}_i^2 = \mathbb{E}\left[\prod_{j=1}^K X_j^2\right] = \prod_{j=1}^K \mathbb{E}[X_j^2] = 1 \tag{25}$$

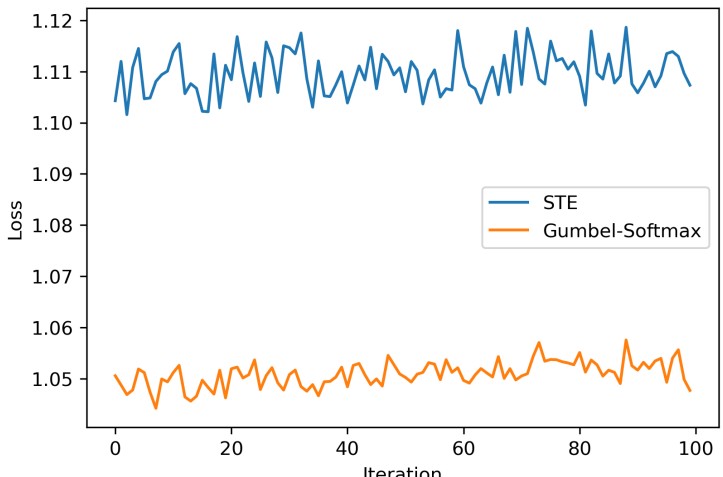

Figure 8: Mean squared error loss for optimization of variational parameters with a Gaussian likelihood using the straight-through gradient estimator and Gumbel-softmax reparameterization trick.

since $\mathbb{E}[X_j^2] = \text{Var}(X_j) + \mathbb{E}[X_j]^2 = 1$ for $X_j \sim \mathcal{N}(0,1)$, $j = 1, \ldots, K$. By the strong law of large numbers (SLLN), $\delta(\mathbf{u}, \mathbf{v}) \to 1$ almost surely as $D \to \infty$. Moreover, by the central limit theorem (CLT), $\delta(\mathbf{u}, \mathbf{v}) \sim \mathcal{N}(1, \sigma_K^2/\sqrt{D})$ as $D \to \infty$ where $\sigma_K^2 = \text{Var}\left(\prod_{j=1}^{K} X_j^2\right)$ for $X_j \sim \mathcal{N}(0,1)$.

Suppose instead there exists some $j^*$ such that $\mathbf{x}_{j^*} \neq \mathbf{y}_{j^*}$. Then as $D \to \infty$,

$$\delta(\mathbf{x}, \mathbf{y}) = \frac{1}{D} \sum_{i=1}^{D} \mathbf{u}_i \mathbf{v}_i = \mathbb{E}\left[\prod_{j=1}^{K} X_j Y_j\right] = \prod_{j=1}^{K} \mathbb{E}[X_j Y_j], \tag{26}$$

for $X_j, Y_j \sim \mathcal{N}(0,1)$ independent across $j = 1, \ldots, K$, which evaluates to zero if $X_{j^*}$ and $Y_{j^*}$ are independent. By SLLN, $\delta(\mathbf{u}, \mathbf{v}) \to 0$ almost surely as $D \to \infty$. Furthermore, in this case $\{\mathbf{u}_i \mathbf{v}_i\}_{i=1}^{D}$ are i.i.d. mean-zero, variance-one random variables, so by the CLT, $\delta(\mathbf{u}, \mathbf{v}) \sim \mathcal{N}(0, 1/\sqrt{D})$ as $D \to \infty$, so $\delta(\mathbf{u}, \mathbf{v}) \to 0$ as $D \to \infty$. $\qquad\square$

## B  VARIATIONAL RESONATOR NETWORK WITH GAUSSIAN LIKELIHOOD

Suppose we have the likelihood

$$p(\mathbf{x}|\mathbf{z}_1, \mathbf{z}_2) = \mathcal{N}(\mathbf{x}; \mathbf{X}_1 \mathbf{z}_1 \odot \mathbf{X}_2 \mathbf{z}_2, I) \tag{27}$$

and variational posterior

$$q_\phi(\mathbf{z}_1, \mathbf{z}_2|\mathbf{x}) = q_{\phi_1}(\mathbf{z}_1|\mathbf{x}) q_{\phi_2}(\mathbf{z}_2|\mathbf{x}) \tag{28}$$

Thus, the variational parameters are $\phi = \{\phi_1, \phi_2\}$ such that $\phi_1, \phi_2$ are logit estimates of the distributions of $\mathbf{z}_1, \mathbf{z}_2$ given $\mathbf{x}$, respectively; i.e. $q_{\phi_1}(\mathbf{z}_1|\mathbf{x}) = \text{softmax}(\phi_1)^\top \mathbf{z}_1$ and $q_{\phi_1}(\mathbf{z}_1|\mathbf{x}) = \text{softmax}(\phi_2)^\top \mathbf{z}_2$. Let $\mathbf{p}_1 = \text{softmax}(\phi_1)$ and $\mathbf{p}_2 = \text{softmax}(\phi_2)$. We have

$$\log q_\phi(\mathbf{z}_1, \mathbf{z}_2|\mathbf{x}) = \log(\mathbf{p}_1^\top \mathbf{z}_1) + \log(\mathbf{p}_2^\top \mathbf{z}_2) \tag{29}$$

$$\log p(\mathbf{x}, \mathbf{z}_1, \mathbf{z}_2) = \log p(\mathbf{x}|\mathbf{z}_1, \mathbf{z}_2) + \log p(\mathbf{z}_1) + \log p(\mathbf{z}_2) \tag{30}$$

We have uniform priors on $\mathbf{z}_1$ and $\mathbf{z}_2$ so we can ignore those terms. Thus we want to maximize the quantity

$$\mathcal{L}(\phi) = -\frac{1}{2} \mathbb{E}_{q_\phi(\mathbf{z}_1, \mathbf{z}_2|\mathbf{x})}[\|\mathbf{x} - \mathbf{X}_1 \mathbf{z}_1 \odot \mathbf{X}_2 \mathbf{z}_2\|^2] + H(\mathbf{p}_1) + H(\mathbf{p}_2) \tag{31}$$

In order to optimize the variational parameters, we have to use techniques such as the straight-through gradient estimator (STE) (Bengio et al., 2013) or the Gumbel-softmax reparameterization trick (Jang et al., 2017). Unfortunately, as shown in Figure 8, this leads to unstable losses empirically and did not result in any meaningful results.

## C  ADDITIONAL DERIVATIONS

We have the loss function given in Eq. 16:

$$\mathcal{L}(\phi) = -\delta(\mathbf{x}, \mathbf{X}_1\mathbf{p}_1 \odot \mathbf{X}_2\mathbf{p}_2) + \sum_{j=1}^{2} D_{\mathrm{KL}}[q_{\phi_j}(\mathbf{z}_j|\mathbf{x})\|p(\mathbf{z}_j)] \tag{32}$$

If we assume $p(\mathbf{z}_1)$ and $p(\mathbf{z}_2)$ are uniform (i.e. $p(\mathbf{z}_j) = 1/n$ where $n$ is the codebook size, for $j = 1, 2$), the KL term evaluates to:

$$D_{\mathrm{KL}}[q_{\phi_j}(\mathbf{z}_j|\mathbf{x})\|p(\mathbf{z}_j)] = \mathbb{E}_{q_{\phi_j}(\mathbf{z}_j|\mathbf{x})}[\log q_{\phi_j}(\mathbf{z}_j|\mathbf{x}) - \log p(\mathbf{z}_j)] \tag{33}$$

$$= \mathbb{E}_{q_{\phi_j}(\mathbf{z}_j|\mathbf{x})}[\log q_{\phi_j}(\mathbf{z}_j|\mathbf{x})] - \mathbb{E}_{q_{\phi_j}(\mathbf{z}_j|\mathbf{x})}[\log(1/n)] \tag{34}$$

$$= -H(q_{\phi_j}(\cdot|\mathbf{x})) + C = -H(\mathbf{p}_j) + C \tag{35}$$

where $\mathbf{p}_j = \mathrm{softmax}(\phi_j)$, $j = 1, 2$. $C$ is a constant which we can ignore. This gives us the loss

$$\mathcal{L}(\phi) = -\delta(\mathbf{x}, \mathbf{X}_1\mathbf{p}_1 \odot \mathbf{X}_2\mathbf{p}_2) - H(\mathbf{p}_1) - H(\mathbf{p}_2) \tag{36}$$

Substituting $\delta(\mathbf{x}, \mathbf{y}) = \mathbf{x}^\top\mathbf{y}/D$, we simplify the loss function using the identity $\mathbf{x}^\top(\mathbf{a} \odot \mathbf{b}) = (\mathbf{x} \odot \mathbf{b})^\top\mathbf{a}$. Let $k \neq j$ be the index of the other codebook. The loss can be rewritten to isolate $\mathbf{p}_j$:

$$\mathcal{L}(\phi) = -\frac{1}{D}(\mathbf{x} \odot \mathbf{X}_k\mathbf{p}_k)^\top\mathbf{X}_j\mathbf{p}_j - H(\mathbf{p}_j) \tag{37}$$

$$= -\mathbf{v}_j^\top\mathbf{p}_j - H(\mathbf{p}_j) \tag{38}$$

where we define the projected similarity vector $\mathbf{v}_j = \frac{1}{D}\mathbf{X}_j^\top(\mathbf{x} \odot \mathbf{X}_k\mathbf{p}_k)$.

The gradient with respect to the probability vector $\mathbf{p}_j$ is:

$$\nabla_{\mathbf{p}_j}\mathcal{L} = -\frac{1}{D}\mathbf{X}_j^\top(\mathbf{x} \odot \mathbf{X}_k\mathbf{p}_k) + \mathbf{1} + \log\mathbf{p}_j \tag{39}$$

## D  VARIATIONAL RESONATOR AUTOENCODER

The general loss for the VRAE is

$$\mathcal{L}(\theta, \psi, \phi) = \mathbb{E}_{q_\psi(\mathbf{x}|\mathbf{y})}[\log q_\psi(\mathbf{x}|\mathbf{y}) - \log p_\theta(\mathbf{y}|\mathbf{x})]$$
$$+ \mathbb{E}_{q_\psi(\mathbf{x}|\mathbf{y})}\mathbb{E}_{q_\phi(\mathbf{z}|\mathbf{x})}[\log q_\phi(\mathbf{z}|\mathbf{x}) - \log p(\mathbf{x}|\mathbf{z})] \tag{40}$$

Let us consider a specific case for image generation where $p_\theta(\mathbf{y}|\mathbf{x}) = \mathcal{N}(\mathbf{y}; g_\theta^{\mathrm{dec}}(\mathbf{x}), I)$. Then the resulting loss function is

$$\mathcal{L}(\theta, \psi, \phi) = \|\mathbf{y} - g_\theta^{\mathrm{dec}}(\mathbf{x})\|^2 - \delta(\mathbf{x}, \hat{\mathbf{x}}_\lambda(\mathbf{x})) - \sum_{j=1}^{K} H[p_\lambda(\mathbf{x})_j] \tag{41}$$

where $\mathbf{x} = f_\psi^{\mathrm{enc}}(\mathbf{y})$, $p_\lambda(\mathbf{x})_j = \mathrm{softmax}(h_\lambda(\mathbf{x})_j)$, and $\hat{\mathbf{x}}_\lambda(\mathbf{x}) = \bigodot_{j=1}^{K} \mathbf{X}_j p_\lambda(\mathbf{x})_j$.

## E  DECOMPOSITION OF CORRELATED REAL VECTORS

Figure 9 compares VRN against resonator network for varying levels of correlation.

## F  EXPERIMENTAL DETAILS

**VRN Hyperparameters**  We use both SGD and Adam optimizers for the VRN inference process. We run the optimizer for 100 iterations with learning rate 0.02. We consider different levels of entropy regularization $\gamma \in \{-1, -0.1, 0, 0.1, 1\}$. The results for bipolar codebooks use Adam with entropy regularization constant $\gamma = 1$, while real codebooks use Adam with $\gamma = 0$.

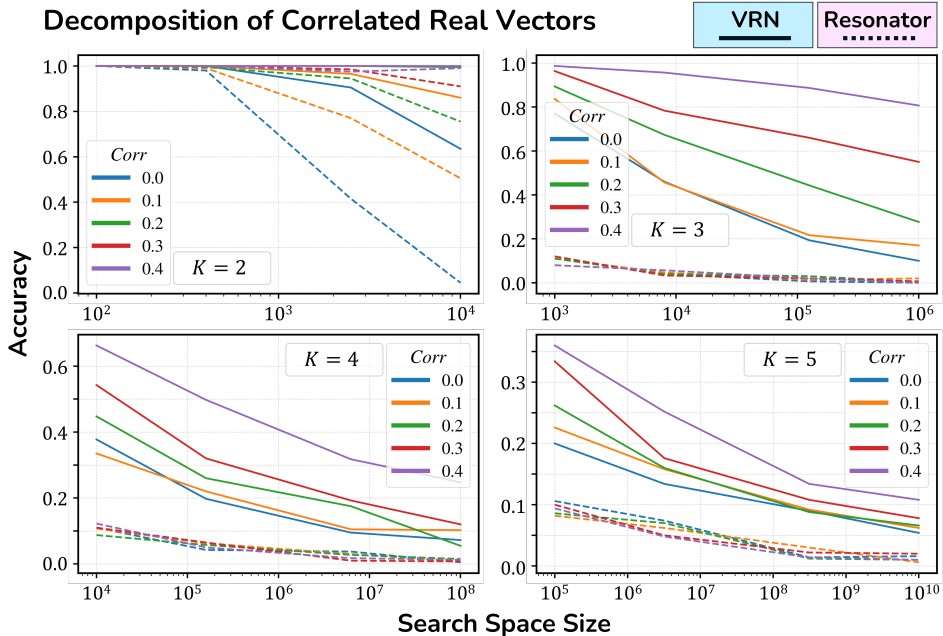

Figure 9: Decomposition accuracy of resonator network (dotted line) and VRN (solid line) over varying search space sizes $n^K$ for real codebooks. Different line colors correspond to levels of correlation. Each plot corresponds to a different number of codebooks ($K = 2, 3, 4, 5$).

**Resonator Network Hyperparameters** We run the resonator network for 100 iterations.

# G ABLATIONS

## G.1 EFFECT OF ENTROPY REGULARIZATION

We consider different levels of entropy regularization $\gamma \in \{-1, -0.1, 0, 0.1, 1\}$. While theory suggests $\gamma > 0$, we include $\gamma \leq 0$ for completeness. See Figure 10 for a comparison.

## G.2 EFFECT OF OPTIMIZER

We consider stochastic gradient descent (SGD) and Adam (Kingma & Ba, 2017) optimizers for the optimization of the loss function given in Eq. 17. SGD results in gradients similar to the resonator network update rule (Eq. 18). See Figure 10 for a comparison.

## G.3 CHOICE OF ACTIVATION FOR RESONATOR NETWORK ON REAL VECTORS

We consider two activations, $\tanh(x)$ and $\mathrm{norm}(x) = x/\|x\|$. See Figure 11 for a comparison. $\tanh$ generally outperforms $\mathrm{norm}$.

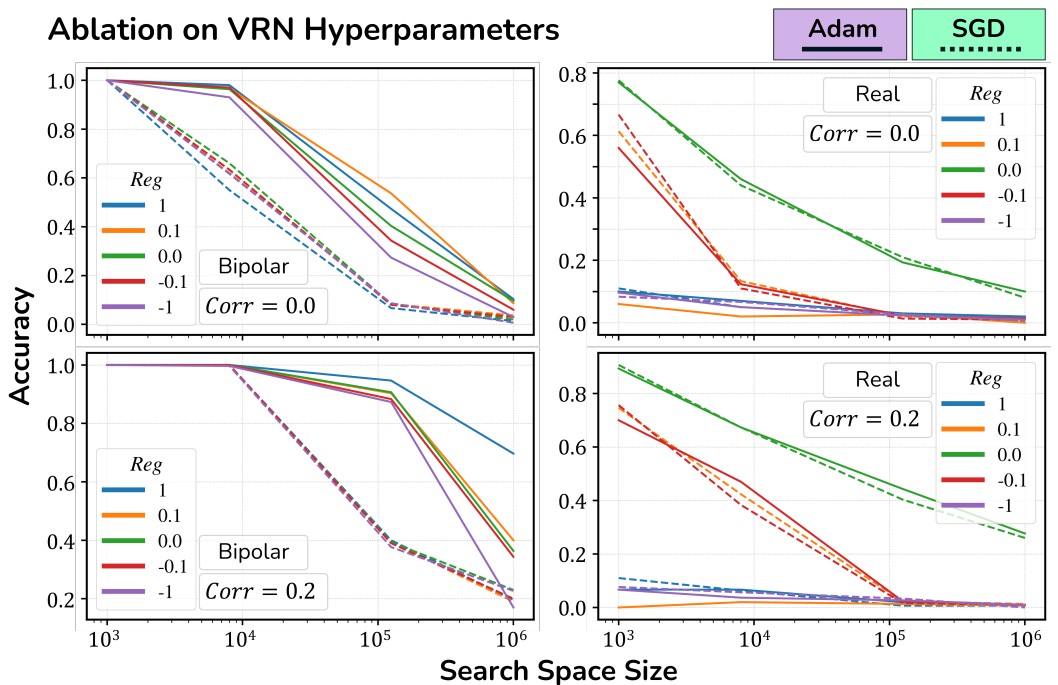

Figure 10: Comparison between SGD (dotted) and Adam (solid) for different levels of entropy regularization. Each plot corresponds to a different degree of correlation and real/bipolar combination. The number of codebooks is set to $K = 3$, and vectors are of dimension $D = 1000$.

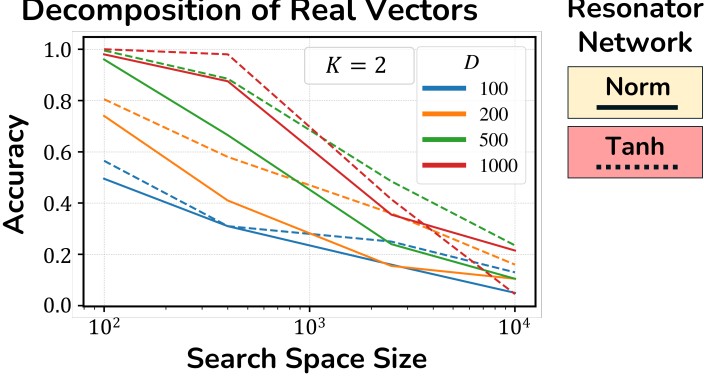

Figure 11: Comparison between tanh (dotted line) and norm (solid line) activations for the resonator network on real vectors.

