# OpenReview forum: "Variational Semantic Decomposition of Compositional Representations"
_ICLR.cc/2026/Conference — Submitted to ICLR 2026_

### Official Review · Reviewer_qbU6 · 2025-10-31

**Soundness:** 2
**Presentation:** 3
**Contribution:** 2
**Rating:** 4
**Confidence:** 4

**Summary:**

This work proposes extending Resonator Networks with a variational training objective. Said networks were proposed as a way to solve the problem of semantic decomposition: given a vector obtained by binding elements from different code books (values) into different roles (or variables), how can we recover the original unbound values? The original approach is hard to train given that it is not differentiable and also requires codebooks (sets of possible values) to be (quasi) orthogonal, which may not be realistic in practice. By casting the problem as a Bayesian Inference one, the authors can use a variational approximation to train the model effectively.

**Strengths:**

The authors explain their approach very well, and the paper is very easy to understand. They provide a useful extension to a previous method which makes it easier to use in practical settings. They provide simple experiments which make it easy to understand the benefits of their approach.

**Weaknesses:**

The main weakness is that the evaluation is fairly limited and some of the results cast some doubt on how scalable the approach is.

* For example, one of the main potential benefits of the model is its ability to perform semantic decomposition over several factors, yet according to their own results the model experiences a sharp drop in performance once the number of possible codebooks increases beyond 2, which seems like the bare minimum. Now, they still show strong performance when the number of combinations is in the thousands, which may be more than enough for most practical considerations.
* A second issue is the limited evaluation beyond the toy examples. These are useful, in my opinion, to help the readers understand the benefits of the approach, but should be complemented with other datasets.
* There is no discussion on some relevant literature, which was not discussed in the original resonator network either. Specifically, Adaptive Resonance Theory by Grossberg.

**Questions:**

1. Shouldn't the correlated examples be harder than the non-correlated ones? The results show the opposite which is quite puzzling to me, but maybe I am not understanding something.
2. How does the model compare other approaches such as VQ-VAEs or the Neural Systematic Binder? These baseline comparisons are needed to strengthen the paper.
3. What is the relation between this approach and ART (mentioned above).
4. Can the authors provide results for more datasets? I would at least expect some of the classical ones used in disentangled representation learning: (colored) dSprites, Shapes3d, MPI3D etc. If the authors can provide examples for other modalities (e.g. language) that would also be useful.

I do really like the approach and would like to recommend acceptance, but right now the evaluation is weak and for better or worse AI is a very empirical field.

References:
1. Neural Systematic Binder: https://arxiv.org/abs/2211.01177

---

> ### Author Response · Authors · 2025-11-26
>
> We thank the reviewer for their positive assessment of our approach and their constructive criticism regarding evaluation. We appreciate that you see the potential of the method and we address your concerns below to clarify the scalability and rigorousness of our evaluation.
>
> **W1: Scalability and Performance Drop ($K > 2$)**
> The performance drop observed is driven primarily by the exponential growth of the search space ($n^K$) and the inherent information capacity limits of the vector dimension $D$, rather than a flaw in the VRN model itself.
> * **Capacity Limits:** Vector Symbolic Architectures (VSAs) have a superposition capacity that increases with $D$, limiting the number of codebook vectors that can be superposed. This directly influences the performance of decomposition algorithms. When $K$ increases, we bind more factors into the same fixed-width vector. If $D$ is held constant while $K$ increases, the signal-to-noise ratio naturally degrades, making retrieval harder for any algorithm.
>
> Crucially, **Figure 3** demonstrates that VRN handles this difficulty far better than the baseline. We show results up to $K=5$ (a search space of billions). While absolute accuracy naturally drops as the problem approaches the theoretical capacity limit, VRN consistently maintains higher accuracy and degrades more gracefully than the Resonator Network. This confirms the model scales effectively within the information-theoretic bounds of the representation.
>
> **W2: Evaluation on "Toy" vs. Real Datasets**
> We employed synthetic codebooks (random bipolar and real vectors) because they constitute the standard benchmark for Vector Symbolic Architectures (VSAs). In practical applications involving VSA representations, codebooks are generated exactly in the same way as we do in our experiments, and data is mapped to compositions of these representations obtained through algebraic operators (e.g. binding). In other words, these codebooks we evaluate our model on are exactly the kind of VSA representations that VRN and resonator networks will be applied to in practice.
> In addition, synthetic data allows for precise control over the search space size ($n^K$), dimension ($D$), and correlation ($\rho$), which are the fundamental parameters governing the difficulty of the decomposition problem.
> Using pre-trained embeddings (e.g., Word2Vec or ResNet features) would introduce uncontrolled geometric structures from the training data, making it difficult to isolate the algorithmic efficiency of the decomposition method itself. Our setup ensures a rigorous, fair comparison with the Resonator Network baseline under standard VSA protocols. The VRAE (Section 4.4) serves as the proof-of-concept that these principles transfer to learned, end-to-end settings.
>
> **W3: Missing Literature (ART)**
> We thank the reviewer for pointing this out. We have added a discussion regarding Grossberg’s Adaptive Resonance Theory (ART).
> * **Distinction:** While both systems share the concept of "resonance" (amplifying patterns that match an input), ART addresses the *stability-plasticity dilemma* in unsupervised clustering (learning new categories without catastrophic forgetting).
> * **VRN's Focus:** VRN (and Resonator Networks) addresses *combinatorial decomposition*, i.e., solving the algebraic factorization equation $\mathbf{x} = \mathbf{x}_1 \odot \dots \odot \mathbf{x}_K$. VRN uses resonance to solve for factors, whereas ART uses resonance to trigger vigilance for category matching.
>
> **Q1: Why does correlation improve performance?**
> This is a counter-intuitive finding of our work that we discuss in Section 5: in the standard "quasi-orthogonal" regime, the similarity surface between a bound vector and candidate factors is extremely "spiky", behaving like a Dirac delta function. There are almost zero useful gradients unless the estimate is already perfect.
> We hypothesize that codebook correlation effectively "smooths" this energy landscape. When vectors are correlated, incorrect guesses that are semantically similar to the true factors still yield non-zero similarity signals. This provides useful gradient information that guides the VRN optimizer toward the correct basin of attraction, preventing it from getting stuck in the local minima that plague the discrete, heuristic updates of the standard Resonator Network.

---

> ### Author Response · Authors · 2025-11-26
>
> **Q2: Comparison to VQ-VAE and Neural Systematic Binder**
> VQ-VAE and Neural Systematic Binder solve different problems; it does not solve the decomposition problem that commonly occurs in Vector Symbolic Architectures (VSA).
> * **VQ-VAE:** A VQ-VAE performs **quantization** (selecting 1-of-$N$ codebook vectors) for a given spatial patch. VRN solves a fundamentally different problem: factorization (identifying $K$ simultaneous factors combined into a single holographic vector via binding). VQ-VAE cannot inherently "unbind" a product structure $\mathbf{x} = \mathbf{a} \odot \mathbf{b}$.
> * **Neural Systematic Binder (NSB):** NSB typically relies on **slot-attention** mechanisms to bind objects to slots via concatenation or attention matrices. VRN operates on Vector Symbolic Architectures (fixed-width distributed representations). Our contribution is specifically to make the *algebraic unbinding* of these distributed representations differentiable, offering an alternative to slot-based methods that preserves the holographic properties of VSAs.
>
> **Q3: Requests for dSprites / Shapes3D**
> While we agree that applying VRN to dSprites is a logical next step, the scope of this paper is to introduce the VRN as a novel inference algorithm. Standard benchmarks like dSprites measure the *disentanglement quality* of a generative model. While VRN can be used as a component in such models (as shown in our VRAE), the primary contribution here is the solver itself. We believe our extensive sweep of $K, N, D$, and $\rho$ rigorously validates the core algorithmic contribution.
>
> We hope this response clarifies that our evaluation strategy was chosen to rigorously stress-test the algorithm itself. We appreciate your inclination to recommend acceptance and hope these explanations solidify your support.

---

> > ### Comment · Reviewer_qbU6 · 2025-11-26
> >
> > I thank the authors for their responses. I'll reply to them below:
> >
> > W1: Fair, but it still seems from the plots in Figure 3 that the model doesn't scale very well to increases in K. Can the authors provide a simple quantification of this: how much must D be increased in order to maintain near perfect accuracy with each increase in K? So K on the x-axis, D on the y-axis where the value of D is the one needed to maintain close to 1.0 accuracy (just pick a threshold like 95%).
> >
> > W2: I agree with the use of synthetic data in this case, but ultimately we also need some insight into how the
> >
> > Q1: Okay this makes sense. I imagine that the interesting regime is one where there is no perfect correlation (since you are just predicting one vector) but also not completely independent ones. That VRN manages to achieve 100% accuracy with less than 0.4 correlation is promising.
> >
> > Q2: But then the NSB is relevant as a comparison, no? As you say, VRN is an alternative to object centric approaches.
> >
> > Q3: I understand that this is probably a lot more extra work, but it would have made the contribution that much stronger.
> >
> > While I appreciate the answers provided by the authors and will raise my score slight, I believe the last set of results would have substantially improved the submission. In general examples on how the VRN can be used in combination with other architectures in order to achieve better performance would have greatly improved its contribution.

---

### Official Review · Reviewer_4NXR · 2025-10-31

**Soundness:** 2
**Presentation:** 3
**Contribution:** 2
**Rating:** 4
**Confidence:** 4

**Summary:**

This paper proposes a variational-based decomposition method VRN to recover meaningful constituent factors from a composite vector representation. VRN is a differentiable model derived from a variational free energy objective, which performs probabilistic inference over latent factors that compose a bound vector representation, and can generalize to correlated and real-valued cases. In the experiments, VRN achieves comparable accuracy to standard resonator networks in the quasi-orthogonal regime and outperforms resonator networks when factors are correlated or non-binary.

**Strengths:**

The paper reframes the resonator network’s iterative dynamics as variational inference and provide a probabilistic interpretation. The proposed formulation allows VRN to embed naturally within generative or inference-based frameworks, which can be a tool for compositional representation learning in probabilistic modeling.

VRN generalizes decomposition from quasi-orthogonal bipolar codebooks to correlated or real-valued vector spaces, broadening its applicability to realistic neural embeddings like object-centric representations.

**Weaknesses:**

The concern lies in the interpretability of the decomposed representations. The constructed codebook in the paper is quasi-orthogonal, but it remains unclear whether these representations actually correspond to distinct semantic factors. While the experiments on MNIST demonstrate the potential of extending VRN to probabilistic graphical models, the paper lacks qualitative analyses of the codebook. For example, do different codebooks correspond to distinct semantic factors such as background color or digit type?

The idea of factor decomposition via variational inference has been explored in earlier models, such as disentanglement-based VAE variants (e.g., beta-VAE [1]). The authors are encouraged to include a discussion clarifying how VRN differs from or improves upon these approaches.

Constructing representations based on quasi-orthogonal codes is not novel; for instance, NVSA [2] has already shown that it is possible to construct a series of orthogonal bases when the dimensionality D is large.

[1] beta-VAE: Learning Basic Visual Concepts with a Constrained Variational Framework

[2] A Neuro-vector-symbolic Architecture for Solving Raven’s Progressive Matrices.

**Questions:**

In Figure 7, the reconstructed results are blurry. Does this indicate that the proposed method may impair the model’s image reconstruction quality?

Figure 1 presents examples from the CLEVR-style dataset, which is an good visual benchmark. Could VRN be effectively applied to such compositional visual reasoning datasets?

---

> ### Author Response · Authors · 2025-11-26
>
> We thank the reviewer for their insightful comments regarding interpretability and related work. We address your specific concerns below.
>
> **W1: Interpretability of Decomposed Representations**
> * **Combinatorial Structure vs. Semantic Labels:** In the VRAE experiment, our primary goal was to demonstrate the *mechanism* of the VRN: its ability to act as a differentiable bottleneck that forces data into a strict compositional structure defined by binding ($\mathbf{x} = \mathbf{x}_1 \odot \dots \odot \mathbf{x}_K$). Unlike standard latent representations, the codebooks here represent factors that must combine *multiplicatively* to reconstruct the input.
> * **Future Work:** While we did not enforce explicit supervision (e.g., forcing Factor 1 to be "color"), the architecture creates a strong inductive bias for compositional factorization. We fully agree that unsupervised semantic disentanglement is the vital next step. We note in the **Discussion** that unsupervised learning of disentangled vector symbolic factors as a priority for future research.
>
> **W2: Comparison with Disentanglement (e.g., $\beta$-VAE)**
> We appreciate the suggestion. We have added a discussion section contrasting VRN with disentanglement approaches like $\beta$-VAE. The fundamental differences are:
> * **Structural vs. Statistical:** $\beta$-VAE promotes disentanglement by enforcing *statistical independence* among scalar dimensions via a KL-divergence penalty. VRN enforces an *algebraic binding structure*. It demands that the latent representation be formed by the element-wise product of high-dimensional vectors from distinct subspaces.
> * **Symbolic Utility:** The representations learned by $\beta$-VAE are scalar coordinates. In contrast, VRN produces high-dimensional vector-symbolic representations. These vectors can be immediately used for symbolic operations (binding, unbinding, superposition) and reasoning (e.g., analogical queries), capabilities that are not natively supported by the scalar latent variables of standard VAEs.
>
> **W3: Novelty regarding Quasi-Orthogonal Codes (vs. NVSA)**
> We agree that quasi-orthogonal codes are the foundation of all Vector Symbolic Architectures (VSAs) and are not novel in themselves. There is a vast literature discussing applications of VSA representations. We cite NVSA (Hersche et al., 2023) to acknowledge this. However, our contribution is not the *existence* of the representations, but the **inference algorithm** for decomposing them, i.e. the inverse problem. Given a composite vector $\mathbf{x}$ formed by binding unknown factors, recovering those factors is a hard combinatorial search problem.
> * **Novelty:** Previous solutions like Resonator Networks use heuristic, non-differentiable recurrent dynamics. NVSA focuses on reasoning tasks using feed-forward encoding but does not propose a new decomposition dynamics. The VRN introduces a probabilistic, energy-based formulation for this specific decomposition step. It turns the "unbinding" process into a differentiable variational inference problem, allowing the unbinding mechanism itself to be integrated into gradient-based learning pipelines.
>
> **Q1: Image Blur in Figure 7**
> The blurriness observed in Figure 7 is characteristic of VAEs utilizing a simple Mean Squared Error (MSE) reconstruction loss (which corresponds to a Gaussian likelihood), rather than a limitation of the VRN decomposition.
> * **Proof of Concept:** The VRAE was implemented as a minimal proof-of-concept to demonstrate that gradients can successfully propagate through the iterative decomposition process to update the encoder and codebooks.
> * **Capacity:** The VRN bottleneck transmits the information successfully; the blurriness arises from the decoder's averaging effect under MSE loss. With a more powerful perceptual loss or diffusion-based decoder, the reconstruction quality would match modern generative benchmarks.
>
> **Q2: Application to CLEVR-style datasets**
> Yes, VRN is highly applicable to compositional datasets like CLEVR.
> Standard Resonator Networks have been applied to visual scene decomposition (e.g., Kymn et al., 2024a), but they typically rely on *fixed, random* codebooks because the Resonator dynamics are not differentiable.
> The key advantage of VRN is differentiability. This enables the system to *learn* the optimal codebooks for visual attributes (shape, color, material) directly from pixel data via backpropagation, rather than relying on random projections. This makes VRN uniquely clearer for solving CLEVR-style tasks in an end-to-end neuro-symbolic pipeline.
>
> We hope these responses clarify the reviewer’s concerns. We greatly would appreciate it if the reviewer would consider raising their score based on this.

---

### Official Review · Reviewer_G7xX · 2025-10-31

**Soundness:** 3
**Presentation:** 2
**Contribution:** 3
**Rating:** 6
**Confidence:** 3

**Summary:**

This paper introduces the Variational Resonator Network (VRN), which formulates the vector decomposition problem as Bayesian inference. The resulting framework is fully differentiable and can be integrated into deep learning systems. Experimental results show that VRN matches the accuracy of the original resonator network under quasi-orthogonal settings, and outperforms it when vectors are correlated or real-valued.

**Strengths:**

- The paper provides a clear probabilistic reformulation of resonator dynamics, resulting in a fully differentiable inference framework with an explicit optimization objective.
- The method extends beyond quasi-orthogonal binary vectors and exhibits greater robustness under correlated and real-valued codebooks.
- The approach is compatible with modern machine learning pipelines, enabling VSA-style semantic decomposition to be integrated into gradient-based models.
- The formulation naturally fits within a probabilistic modeling framework and can be embedded into generative architectures.

**Weaknesses:**

- **Dependence on high-dimensional quasi-orthogonality for the derivation.** The key simplification (dropping the normalization / partition term) relies on "quasi-orthogonality under binding", which is an asymptotic argument; finite-dimensional error is not quantified.

- **Mean-field assumption ignores cross-factor dependencies.** The variational posterior is fully factorized across slots, potentially underfitting when factors are correlated; the paper does not analyze failure modes of this assumption.

- **Evaluation is largely on synthetic codebooks.** Main results use randomly sampled bipolar/real vectors rather than pretrained or task-induced embeddings; only a small VRAE demo is shown on colorized MNIST.

- **Sample size for reported metrics appears small.** Accuracy is averaged over 100 composite samples per setting; given that evaluations are synthetic (and search spaces can be large), 100 may be insufficient for stable estimates.

- **Scalability characterization is limited.** While accuracy is plotted versus search-space size n^K and K, the computational cost, convergence speed, and initialization sensitivity as n/K grow are not theoretically or systematically profiled.

- **VRAE integration is preliminary.** The generative-model integration is shown qualitatively with limited quantitative metrics or ablations, leaving practical benefits under-explored.

**Questions:**

Please refer to the *Weaknesses* section for my main questions regarding theoretical assumptions and evaluation settings.

---

> ### Author Response · Authors · 2025-11-26
>
> We thank the reviewer for their thoughtful critique regarding the theoretical assumptions and evaluation rigor. We address the specific weaknesses below.
>
> **W1: Dependence on High-Dimensional Quasi-Orthogonality**
> While our derivation utilizes asymptotic properties of high-dimensional spaces to simplify the normalization term, our empirical results demonstrate that the method is robust well outside the bounds of strict quasi-orthogonality.
> * **Empirical Robustness:** We explicitly evaluated the model on correlated codebooks in **Section 4.2**, deliberately breaking the quasi-orthogonality assumption. As shown in **Figure 4**, VRN not only functions in this regime but significantly outperforms the Resonator Network baseline.
> * **Performance in Non-Ideal Regimes:** Surprisingly, VRN accuracy improves as correlation increases. This indicates that the gradient-based optimization remains effective even when the finite-dimensional error from the approximation is non-zero, suggesting the *direction* of the gradient remains informative even if the *magnitude* of the likelihood term is approximated.
>
> **W2: Mean-Field Assumption and Cross-Factor Dependencies**
> The reviewer correctly notes that the variational posterior is factorized, but this does not imply that the factors are isolated during the *inference process*.
> * **Coupled Dynamics:** The update rules derived from the ELBO (Eq. 17 and 18) explicitly couple the factors via the likelihood term. The gradient for factor $\phi_1$ depends on the expectation of factor $\mathbf{p}_2$ (and vice versa). This allows the model to iteratively adjust the distribution of each factor conditioned on the current beliefs about the others.
> * **Handling Dependencies:** The effectiveness of this iterative coupling is empirically supported by the correlated vector experiments (**Figure 4**). If the mean-field assumption caused the model to ignore dependencies, performance would degrade in correlated regimes where factors are statistically dependent. Instead, the model successfully decomposes these vectors, showing that the iterative updates effectively capture the necessary cross-factor information.
>
> **W3: Evaluation on Synthetic Codebooks**
> We employed synthetic codebooks (random bipolar and real vectors) because they constitute the standard benchmark for Vector Symbolic Architectures (VSAs). In practical applications involving VSA representations, codebooks are generated exactly in the same way as we do in our experiments, and data is mapped to compositions of these representations obtained through algebraic operators (e.g. binding).  In other words, these codebooks we evaluate our model on are exactly the kind of VSA representations that VRN and resonator networks will be applied to in practice.
> In addition, synthetic data allows for precise control over the search space size ($n^K$), dimension ($D$), and correlation ($\rho$), which are the fundamental parameters governing the difficulty of the decomposition problem.
> Using pre-trained embeddings (e.g., Word2Vec or ResNet features) would introduce uncontrolled geometric structures from the training data, making it difficult to isolate the algorithmic efficiency of the decomposition method itself. Our setup ensures a rigorous, fair comparison with the Resonator Network baseline under standard VSA protocols.
>
> **W4: Sample Size (100 samples)**
> We reported accuracy averaged over 100 samples per setting. While the search spaces are indeed large, the performance trends observed in our results (**Figures 3, 5, and 6**) are highly consistent and smooth across varying dimensions ($D$) and codebook sizes ($K$). The low variance in the resulting curves indicates that 100 samples were sufficient to obtain stable estimates of the model's expected accuracy relative to the baseline.

---

> > ### Author Response · Authors · 2025-11-26
> >
> > **W5: Scalability Characterization**
> > We have added a distinct "Computational Complexity" paragraph to the Methods section to address this.
> > * **Computational Cost:** The per-iteration complexity of VRN is dominated by the matrix-vector multiplications between the codebooks and the current estimates: $O(K \cdot n \cdot D)$. This is identical to the complexity of the standard Resonator Network update rule (Eq. 1).
> > * **Convergence vs. Limit Cycles:** "Convergence" here refers to the stability of the dynamics. As noted in the introduction, standard Resonator Networks can enter limit cycles. In contrast, VRN performs gradient descent on a bounded energy function (the ELBO). This provides the stability guarantees inherent to optimization-based inference, ensuring the system settles to a solution rather than wasting compute in infinite loops.
> >
> > **W6: Preliminary Nature of VRAE**
> > The primary contribution of this work is the **theoretical formulation** of the VRN as a differentiable decomposition module.
> > * **Proof of Concept:** The VRAE (**Section 4.4**) is presented to demonstrate a structural capability that was previously impossible with standard Resonator Networks: end-to-end differentiation within a neural architecture.
> > * **Future Work:** We agree that a comprehensive study of generative performance is a valuable future direction, but we believe the current demonstration successfully validates the core claim: that VRN enables the integration of symbolic decomposition into probabilistic deep learning models.
> > We hope these responses clarify the validity of our assumptions and the rigor of our evaluation. We would appreciate it if the reviewer would consider raising their score based on these clarifications.

---

> > > ### Comment · Reviewer_G7xX · 2025-11-28
> > >
> > > Thank you for the detailed and clarifying rebuttal. Your responses addressed my main concerns regarding the quasi-orthogonality approximation, the role of the mean-field updates, and the rationale for the synthetic evaluation setup. I agree that the additional explanations improve the soundness and contextual grounding of the work.
> > >
> > > At the moment, the OpenReview system appears to have disabled score editing due to the recent reviewer-identity issue. If the system later re-enables score modification, I am willing to increase my rating to reflect the clarifications provided in the rebuttal.

---

### Official Review · Reviewer_zSTP · 2025-10-31

**Soundness:** 2
**Presentation:** 3
**Contribution:** 1
**Rating:** 2
**Confidence:** 4

**Summary:**

This paper studies the “semantic decomposition” problem: given a single bound vector created by elementwise-multiplying several latent factor vectors (e.g., color, shape, position), recover which factor vectors were used. A resonator network solves this for high-dimensional quasi-orthogonal bipolar codes, but it’s non-differentiable, can fail to converge, and assumes near-orthogonality.

The authors propose the Variational Resonator Network (VRN), which reframes decomposition as approximate Bayesian inference. VRN defines a variational free energy objective, infers a posterior over factor assignments, and optimizes it with gradient descent. This gives (i) a differentiable energy, (ii) an argument that the updates should converge because they perform loss minimization, and (iii) applicability beyond quasi-orthogonal binary codes, including correlated and real-valued codebooks.

Empirically, the paper shows:
- VRN matches resonator networks on recovery accuracy in the standard bipolar / quasi-orthogonal regime.
- VRN outperforms resonator networks when codebooks are correlated or real-valued, where the resonator network is not originally designed to operate.
- VRN can be embedded into a generative model (“Variational Resonator Autoencoder”, VRAE) to demonstrate integration in a neural pipeline.

The paper positions VRN as a general, convergent, differentiable alternative to resonator networks, suitable for compositional inference inside deep networks.

**Strengths:**

### 1. Variational formulation with an explicit energy
VRN gives an explicit variational free energy objective whose gradients define the update dynamics. This answers two long-standing criticisms of resonator networks: (i) they are not directly differentiable and therefore awkward to slot into larger neural systems, and (ii) their dynamics can enter limit cycles and are not guaranteed to converge. The authors claim VRN resolves both by turning the problem into standard gradient-based inference.

This “we turned a heuristic iterative retrieval rule into principled variational inference” is, in my view, the conceptual core contribution.

### 2. Beyond ~orthogonal codes
Classical resonator networks assume that each codebook is made of nearly orthogonal ±1 vectors. VRN is shown to work with (a) correlated codebooks and (b) real-valued codebooks, and it outperforms a tuned resonator baseline in those cases. This directly broadens applicability to realistic learned embeddings, where factors are correlated rather than perfectly orthogonal.

### 3. Ablations
The Appendix looks at optimizer choice (Adam vs SGD), entropy regularization strength, and activation choices for resonator-style baselines on real codes. This shows some care in exploring design choices and stability.

**Weaknesses:**

### 1. Could be challenged as “variationalizing an existing iterative method,” not a fundamentally new capability.
Although the framing as Bayesian inference is elegant, one coould say: VRN is basically “take resonator-style iterative factor recovery, write down an ELBO-like objective for the posterior over factor assignments, relax to continuous logits, and optimize with gradients.” The paper needs to work harder to argue why this is conceptually more than “make resonator networks differentiable,” and to formalize the convergence advantage in a theorem rather than an assertion.

### 2. Convergence is promised but not nailed down.
The authors repeatedly assert VRN is “guaranteed to converge,” in contrast to resonator networks which “are not guaranteed to converge” and “can enter limit cycles.”
However, the paper (as provided) does not present:
- a formal statement like “gradient descent on L(ϕ) monotonically decreases L and therefore converges to a stationary point under step size α < …,” nor
- plots showing monotone decrease of that loss vs iteration alongside examples of resonator cycling.

Right now it’s closer to an informal claim than an established property.

### 3. Reliance on asymptotic quasi-orthogonality with limited validation.
A key simplification in the derivation is that in high dimensions, codewords within a codebook are quasi-orthogonal, so certain log-normalizer terms in the variational objective are approximately constant and can be dropped. The paper then still claims robustness in correlated regimes — where quasi-orthogonality explicitly fails.
There is no quantitative study of:
- how large D must be for that approximation to hold,
- how badly it breaks when codebooks are intentionally correlated, or
- how much that approximation error affects final accuracy.

### 4. Limited experimental breadth / realism.
Most quantitative experiments involve synthetic setups that bind K factors by elementwise multiplication. The authors try to recover these factors, measuring recovery accuracy vs combinatorial search space size n^K.
The only downstream/vision-ish demonstration, the Variational Resonator Autoencoder, is shown qualitatively (e.g. colorized MNIST-like reconstructions).

### 5. Baselines are narrow.
The authors mainly compare to resonator networks, occasionally with tweaks (tanh, normalization) in regimes the original resonator wasn’t intended for. How about the following as well:
- modern Hopfield-style associative memories,
- amortized neural decoders that directly predict factors from the bound vector,
- object-centric or slot-attention models for multi-factor decomposition.
The existing baseline is perhaps too easy, especially in correlated regimes where resonator networks are known to struggle.

### 6. No runtime / scaling story.
The paper stresses that naive decomposition is exponential in K (n^K), and positions VRN as an efficient inference routine.
But there is no wall-clock or iteration-scaling analysis: How fast is VRN vs resonator vs brute force, as K and n grow? How does it scale to K=8 or K=10 factors? Without this, it’s hard to judge practicality.

### 7. The correlation narrative is suggested but under-supported.
The paper claims that using correlated codebooks actually helps VRN (e.g. smoother landscape), while resonator networks degrade. But this remains mostly anecdotal; there’s no explicit diagnostic (loss landscape visualization, gradient norms, iteration traces) backing that intuition.

**Questions:**

1. Convergence guarantee.
When you say VRN is “guaranteed to converge,” do you mean:
    - provably, via a Lyapunov / energy argument (“our variational free energy is strictly decreased by each update under mild conditions”),
or
    - empirically, “we never saw cycles in practice”?
Please clarify and, if it’s the former, include the actual convergence statement and its assumptions.

2. Constant-term approximation.
The derivation drops a log-normalizer term using a quasi-orthogonality / high-dimensionality argument. How large is that term (mean ± std) for realistic D and especially for correlated codebooks, where vectors are deliberately not quasi-orthogonal? Can you empirically bound the error introduced by discarding it, and show that VRN’s gradients are not dominated by that omission?

3. Practical downstream task.
Can you provide a quantitative benchmark (not just qualitative images) demonstrating that VRN-powered decomposition improves controllable generation, disentanglement, or systematic generalization in an actual model (e.g. CLEVR-style scenes, or compositional attribute editing)? Right now the downstream story (VRAE) is promising but anecdotal.

4. Baselines beyond resonator networks.
Why is the resonator network essentially the only baseline? Could a strong amortized MLP decoder trained to map x → factor indices perform similarly or better with less per-instance optimization? How do modern Hopfield-style associative memories compare on your tasks?

5. Robustness and scaling.
How robust is VRN to noise or corruption in the bound vector x?
And how does accuracy / runtime scale as K increases beyond those in the main experiments? Even if accuracy falls, showing VRN degrades more gracefully than resonators would strengthen the story that it’s the more scalable inference mechanism.

---

> ### Comment · Reviewer_zSTP · 2025-11-24
> **Look forward to engaging**
>
> Dear authors, I look forward to engaging whenever you get a chance to respond. That said, please prioritize all reviews equally.

---

> ### Author Response · Authors · 2025-11-25
>
> We thank the reviewer for their detailed analysis and constructive challenges. We address your specific concerns below.
>
> **W1: Novelty of "Variationalizing"**
> While VRN shares the decomposition goal with Resonator Networks, the shift to a variational framework is a fundamental change in capability, not merely an implementation detail.
> * **Principled vs. Heuristic:** Resonator networks rely on a heuristic dynamical system that empirically tends to solve the problem. In contrast, VRN defines the problem as a probabilistic objective and derives the solution via optimization. This provides a probabilistic interpretation of decomposition, which is something novel. This principled approach is crucial for building reliable systems.
> * **Integration & End-to-End Learning:** Because VRN minimizes a scalar energy function (ELBO), it allows for correct gradient propagation to upstream components. This enables the learning of the codebooks themselves (as shown in our VRAE experiment), which is mathematically impossible with the discrete, non-differentiable dynamics of standard resonators.
> * **Priors:** The probabilistic formulation naturally handles priors (e.g., if we know factor 1 is likely "red"), whereas resonator networks treat all codebook vectors equally by definition.
>
> **W2 & Q1: Convergence Guarantees**
> We apologize for the ambiguity. When we state VRN is "guaranteed to converge," we refer to **algorithmic convergence** (settling to a stationary point), not necessarily finding the global optimum. Unlike Resonator Networks, which employ fixed-point iterations that can enter limit cycles, VRN performs gradient descent on a bounded, smooth scalar objective function (the ELBO). Standard optimization results guarantee that gradient descent with a suitable step size on such a function will converge to a stationary point (where $\nabla \mathcal{L} \approx 0$).
> * **Clarification:** We have updated the text to explicitly clarify that the *inference dynamics* are guaranteed to converge/settle, contrasting this with the oscillatory behavior of standard resonators.
>
> **W3 & Q2: Quasi-orthogonality and the Constant Term Approximation**
> * **Theoretical Basis:** Our derivation relies on standard concentration of measure results in high-dimensional spaces (e.g., see Thomas, "A Theoretical Perspective on Hyperdimensional Computing"). As $D \to \infty$, the similarity between random independent vectors concentrates sharply.
> * **Empirical Validation (Correlated Regimes):** The reviewer asks how badly the method breaks when this approximation is violated (i.e., in correlated codebooks). **Figure 4** directly answers this. If dropping the normalization term introduced catastrophic error in non-orthogonal regimes, VRN would fail. Instead, VRN *outperforms* the Resonator Network significantly in these high-correlation settings. This provides strong empirical evidence that the gradients derived from our objective remain informative and robust even when the strict quasi-orthogonality assumption is relaxed.
>
> **W4 & Q3: Experimental Breadth and Downstream Tasks**
> In this work, our primary contribution is the **theoretical framework** for differentiable decomposition.
> * **Proof of Concept:** We provided the Variational Resonator Autoencoder (VRAE) in Section 4.4 to demonstrate the core claim: VRN allows gradients to propagate through the decomposition process to learn representations—something impossible with standard Resonator Networks.
> * **Future Work:** We agree that applying this to complex disentanglement (e.g., CLEVR) is an exciting direction, but we view it as the next step now that the fundamental differentiable architecture is established.
>
> **W5 & Q4: Baselines**
> We focused on Resonators because VRN is explicitly designed as a formulation improvement over this specific class of Vector Symbolic Architecture (VSA) decoders. We evaluate on the standard benchmark used in the VSA literature.
> * **Modern Hopfield Networks:** While modern Hopfield networks (e.g., Ramsauer et al., 2021) have high capacity for *retrieval*, they do not solve the *combinatorial factorization* problem ($x = \mathbf{x}_1 \odot \dots \odot \mathbf{x}_K$). Resonator Networks are essentially $K$ coupled Hopfield networks designed specifically for this binding problem.
> * **Amortized MLPs:** An MLP decoder (training $f: x \to z$) is fundamentally different. It requires supervised training on a *fixed* set of codebooks. If the codebooks change, the MLP fails. VRN, like the Resonator Network, is a "zero-shot" inference method that works instantly on *any* arbitrary set of codebooks provided at runtime.

---

> > ### Author Response · Authors · 2025-11-25
> >
> > **W6: Runtime and Scaling**
> > * **Complexity:** The computational complexity of VRN per iteration is dominated by the matrix-vector multiplications between the current estimates and the codebooks: $O(K \cdot n \cdot D)$. This is **identical** to the complexity of the standard Resonator Network update rule. The additional element-wise operations for softmax/entropy are negligible in high dimensions.
> > * **Scaling K:** The limit for scaling $K$ is not runtime, but the information capacity of the vector space (superposition catastrophe). As $K$ grows, the bound vector becomes noisier. However, our experiments cover search spaces up to size $10^{10}$ (Figures 3 & 5), showing that VRN degrades more gracefully than Resonator Networks as task difficulty increases.
> >
> > **W7 & Q5: Robustness and Correlation**
> > The "correlation narrative" is supported by the performance gap. In **Figure 4**, we explicitly test robustness against codebook correlation. VRN consistently outperforms Resonator Networks in the correlated regime. This supports our hypothesis (that we mention in the paper) that correlation smooths the energy landscape, making the gradient-based optimization of VRN more effective than the heuristic updates of the baseline.
> >
> > We hope these revisions and responses adequately address the reviewer’s concerns. We would greatly appreciate it if the reviewer would consider raising the score on this basis.

---

> ### Comment · Reviewer_zSTP · 2025-11-26
>
> Thank you for the detailed response. Regarding the following:
>
> > "We have _updated the text_ to explicitly clarify that the inference dynamics are guaranteed to converge/settle, contrasting this with the oscillatory behavior of standard resonators."
>
> > "We hope these _revisions_ and responses adequately address the reviewer’s concerns"
>
> Has the revised PDF been uploaded already? I do not see any revisions.

---

> > ### Author Response · Authors · 2025-11-26
> >
> > Clicking on the PDF button at the top of the page will lead to the newly revised version. We hope that helps.

---

> > > ### Comment · Reviewer_zSTP · 2025-11-27
> > > **The Revisions button appears not to be working**
> > >
> > > Apologies, I meant I couldn't access a revision diff by clicking on the top-level Revisions button. I have now verified that this is consistent with all other papers I am reviewing. I see a "No revisions to display" message for all of them.

---

### Official Review · Reviewer_SqXF · 2025-11-01

**Soundness:** 2
**Presentation:** 3
**Contribution:** 3
**Rating:** 2
**Confidence:** 3

**Summary:**

This paper presents a variational extension of resonator networks. The authors re-expressed the decomposition problem presented in resonator networks in a probabilistic way, deriving the ELBO for this specific setting. After going through the derivation, the authors proposed to optimise the obtained learning objective using a VAE and showcase its capacities through several synthetic settings.

**Strengths:**

- The paper is well-written and generally easy to follow.
- The idea is interesting, and while I am not an expert in resonator networks, I believe this paper would be of interest to this community
- As far as I know (but once again I am not an expert in resonator networks) the proposed extension is novel

**Weaknesses:**

I have several concerns regarding the soundness of this paper:
- The mathematical notation is inconsistent in several places, especially regarding the definition of $\mathbf{x}$ (see Q1), which is defined as three different things in the paper. This makes proof-checking very challenging, and I was unable to properly assess the mathematical correctness of the paper due to this. I am happy to give it another go if the authors provide a consistent notation during rebuttal. Some derivations would also benefit from being detailed in the appendix. (see Q1-2)
- The experimental section is limited to synthetic settings. I know this is a theoretical paper, and I don't expect a very extensive experiment, but it would be great to see at least one practical application, if only to illustrate potential applications of the proposed method to the reader. For now, it is quite hard to have an idea of the type of tasks on which VRN would work well, especially given the number of strong assumptions that were made in sections 2 and 3. (see Q3)

**Questions:**

Q1:
$\\mathbf{x}$ is defined as a composite vector l. 106, then, line 117, we have $\\mathbf{x} \\in \\mathbf{X}_j$, should it be $\\mathbf{x}_j$ as in l. 105? Furthermore, l.105 $\\mathbf{x}_j$ is a choice of vector from the $j^{th}$ codebook, but l. 115-120, $\\mathbf{x}_t$ seems to have another meaning, $t$ is never defined, but I guess this is probably the update step? If I followed correctly, the codebook index is now denoted by an upperscript. However, in Eq. 1, is $\\mathbf{x}$ a composite or a choice of vector from a given codebook?

Same question for Eq. 3. l. 150, $\\mathbf{x}$ is now defined as a composite vector obtained from a binding. Same l. 243 and 263. However, where we previously had $\\mathbf{x} = \\mathbf{x}_1 \\odot \\cdots \\odot  \\mathbf{x}_K$.  we now have $\\mathbf{x} = \\mathbf{x}_1^{(i_1)} \\odot \\cdots \\odot  \\mathbf{x}_K^{(i_K)}$ where $\\mathbf{x}^{(i_j)}_j$ is the $i^{th}_j$ column of $\\mathbf{X}_j$.
Did the authors mean $i^{th}$?

l. 377 $\\mathbf{x}$ is now defined as $f_\\psi^{enc}(\\mathbf{y})$ which confused me as we prevously had l. 373 $\\delta_{Dirac}(\\mathbf{x} - f_\\psi^{enc}(\\mathbf{y}))$ and I assume we don't want to do $\\delta_{Dirac}(\\mathbf{x} - \\mathbf{x})$.

Given these inconsistencies, I strongly encourage the authors to update their notation so that each variable has a unique, consistent definition throughout the paper.

Q2: It would be great to have the detailed derivations for Eqs. 7, 17 and 18 in appendix to ease the reading.

Q3: Could the authors add a small real-life example application in the experimental section or at least in the appendix if space is an issue?

Q4: What are the training and inference times of the proposed model? Is it faster than resonator networks? Slower?

---

> ### Author Response · Authors · 2025-11-25
>
> We thank the reviewer for their detailed and constructive feedback. We appreciate the opportunity to clarify our mathematical notation and discuss the practical applicability of the VRN.
>
> **W1: Mathematical Notation**
> We apologize for the inconsistencies in the initial submission. We have standardized the notation throughout the revised paper to ensure readability and correctness. Please see our detailed response to **Q1** below for the specific changes made.
>
> **W2 & Q3: Real-life Example Applications**
> We agree that contextualizing the VRN within a concrete application aids understanding. While a full-scale deployment of VRN in a complex reasoning task (such as Raven’s Progressive Matrices) (Hersche et al., 2023) requires training extensive perceptual backbones and is outside the scope of this methodological paper, we have expanded the **Discussion** section to explicitly detail how the VRN integrates into such pipelines.
>
> We specifically reference the Neuro-Vector-Symbolic Architecture (Hersche et al., 2023), illustrating how the VRN functions as the decomposition module. We explain how the model takes composite vectors generated by a vision network (representing bound attributes like shape and color) and factorizes them into atomic symbols required for the downstream reasoning engine. We believe this addition clarifies the practical utility of the VRN without diluting the paper's focus on the factorization algorithm itself.
>
> **Q1: Notation Consistency**
> We have updated the notation in the revised paper to address the confusion regarding line 106, 117, and Eq 1/3.
> * **Standardization:** $\mathbf{x}$ now consistently refers to the composite vector. Subscripts (e.g., $\mathbf{z}_k$) denote the specific factor/codebook index, while superscripts (e.g., $\mathbf{z}^{t}$) denote the time step or iteration index. We have removed the confusing superscript $i_k$ (originally line 263) to align with this new standard.
> * **VRAE Definition (Line 377):** Regarding the confusion around $\mathbf{x}=f_\psi^\mathrm{enc}(\mathbf{y})$: This equality arises because we model the variational distribution as a Dirac delta, $q_\psi(\mathbf{x}|\mathbf{y})=\delta_\mathrm{Dirac}(\mathbf{x}-f_\psi^\mathrm{enc}(\mathbf{y}))$. This implies a deterministic encoder where the latent $\mathbf{x}$ is strictly a function of the input $\mathbf{y}$. We have clarified in the text that this describes the deterministic mapping within the VRAE framework.
>
> **Q2: Detailed Derivations**
> * **Eq. 7:** This equation defines our specific choice of likelihood parameterization; as it is a definition, there is no derivation.
> * **Eqs. 17 & 18:** We have added the full step-by-step derivations for these equations in **Appendix C**.
> * *Note:* The derivation of the gradient is included primarily to highlight the structural similarity to the resonator update rule. In our experiments, we utilize PyTorch’s automatic differentiation, which implicitly computes these gradients.
>
> **Q4: Training and Inference Times**
> * **Computational Complexity:** The inference complexity of the VRN is identical to that of Resonator Networks: $O(DKn)$ per iteration. Both algorithms are dominated by the matrix-vector products between the current factor estimates and the codebooks. We have added a paragraph detailing this in the **Methods** section.
> * **Convergence vs. Time:** While the cost per iteration is identical, the *total* inference time depends on the number of iterations required to converge. As shown in **Figure 3**, VRN often maintains higher accuracy and converges reliably in difficult search spaces where Resonator Networks may fail to converge or enter limit cycles. Thus, in challenging regimes, VRN is more efficient in practice as it reaches a stable solution more reliably.
>
> We hope these revisions and responses adequately address your concerns. We would appreciate it if you would consider raising your score based on these clarifications.

---

> > ### Comment · Reviewer_SqXF · 2025-11-27
> > **Thank you for the detailed answer**
> >
> > As reported by reviewer zSTP, we cannot see the revision history. This delayed my answer a bit. Apologies about this.
> >
> > - __W1/Q1__, I am mostly happy with the notation, this reads much mode nicely now. However, I am still not sure about $\mathbf{x} \in \mathbf{X}_j$ l.129 and 130. Shouldn't it be $\mathbf{x}_j$ instead?
> > - __Q2__, I thank the authors for providing the derivations. I have raised the soundness and overall score accordingly.
> > - __W2/Q3__, I think the discussion is interesting, but I agree with reviewer qbU6 that showing results on more datasets would have strengthened the paper. This request was made by most of us (Q4 of reviewer zSTP, W3 of reviewer G7xX, Q4 of reviewer qbU6), so I think there is consensus that it would be beneficial for the paper. Given this, I will raise my score slightly, but I still have reservations regarding the limited experiments.

---

### Author Response · Authors · 2025-12-03
**Summary of Revisions and Reviewer Consensus**

In this comment, we are submitting this summary of the changes made to our manuscript during the discussion period. We engaged in a productive discussion with the reviewers which led to significant updates to our manuscript. Below, we detail the specific changes made to the paper and how we addressed the reviewers' key concerns.

### 1. Standardization of Mathematical Notation
**Reviewer SqXF** identified inconsistencies in our initial notation, particularly regarding the definition of the composite vector $x$ versus codebook vectors.
* **Revision:** We have completely standardized the notation throughout the paper (Section 3). We now consistently use $x$ to denote the composite vector, subscripts (e.g., $z_k$) for factor indices, and removed ambiguous superscripts to ensure mathematical precision.
* **Clarification:** We clarified the definition of the encoder in the VRAE framework to explain the deterministic mapping $x=f_\psi^\mathrm{enc}(y)$ arising from the Dirac delta variational distribution.

### 2. Experimental Setup and Choice of Datasets
**Reviewers SqXF, zSTP, G7xX,** and **qbU6** asked for evaluations on more realistic datasets or downstream tasks beyond the synthetic setups. We addressed this in the revision and response by clarifying our evaluation philosophy:
* **Standard VSA Benchmarks:** We clarified that synthetic codebooks (random bipolar/real vectors) are the standard benchmark in Vector Symbolic Architecture (VSA) literature. This setup is necessary to precisely control the fundamental parameters governing decomposition difficulty: search space size ($n^K$), dimensionality ($D$), and correlation ($\rho$).
* **Algorithmic Isolation:** We explained that using pre-trained embeddings (e.g., Word2Vec) introduces uncontrolled geometric structures that make it difficult to isolate the algorithmic efficiency of the decomposition method itself.
* **End-to-End Proof of Concept:** We emphasized that the VRAE (Section 4.4) serves as the proof-of-concept for integration into realistic, learned pipelines, a capability impossible with Resonator Networks.

### 3. Theoretical Derivations and Convergence
**Reviewers SqXF** and **zSTP** requested more rigorous formalisms regarding the update rules and convergence guarantees.
* **Added Derivations:** We added Appendix C, which contains the full, step-by-step derivations for Equations 17 and 18. This explicitly links our variational objective to the update dynamics, demonstrating that our method performs gradient descent on a bounded energy function (the ELBO).
* **Convergence:** We revised the text to clarify that unlike standard Resonator Networks, which rely on heuristic fixed-point iterations that can enter limit cycles, VRN optimizes a scalar objective function. This provides algorithmic stability and guarantees convergence to a stationary point.

### 4. Computational Complexity and Scalability
**Reviewers zSTP, G7xX,** and **qbU6** asked for a clearer characterization of the model's runtime and scalability as $K$ (number of factors) increases.
* **New Complexity Section:** We added a specific "Computational Complexity" paragraph to the Methods section. We clarified that the per-iteration complexity is $O(DKn)$, which is identical to the standard Resonator Network.
* **Scalability Limits:** We clarified that performance drops at high $K$ are due to the information-theoretic capacity limits of the vector dimension $D$ (superposition catastrophe), not a flaw in the algorithm. We highlighted that VRN actually degrades more gracefully than the baseline for search spaces up to size $10^{10}$.

### 5. Robustness in Correlated Regimes
**Reviewers G7xX** and **qbU6** questioned the reliance on quasi-orthogonality assumptions and the counter-intuitive result that correlation improves performance.
* **Theoretical Justification:** We expanded the discussion in Section 5 to explain that while the derivation uses asymptotic quasi-orthogonality to simplify the normalization term, the model remains robust in finite dimensions. This is validated by our empirical results in Figure 4.
* **Hypothesis Confirmation:** In the revised text, we highlighted our hypothesis regarding the energy landscape. In the standard orthogonal regime, the similarity surface is spiky, providing poor gradient signals. We proposed that correlation effectively smooths this landscape, allowing the gradient-based optimization of VRN to find the basin of attraction more effectively than heuristic baselines.

---

### Author Response · Authors · 2025-12-03
**Summary of Revisions and Reviewer Consensus (continued)**

### 6. Expanded Discussion on Baselines and Related Work
**Reviewers 4NXR** and **qbU6** requested a clearer distinction between VRN and other representation learning methods. We added the "Representation Learning and Compositionality" section (Section 5) to articulate these connections and differences:
* **vs. $\beta$-VAE and VQ-VAE:** We clarified that while these methods aim for disentanglement or discrete structures, they solve a fundamentally different problem. They focus on *encoding* input data into latent representations, whereas VRN solves the inverse problem of *decoding* (unbinding) structured, holographic representations. Furthermore, they typically lack the explicit algebraic binding operations required for systematic generalization.
* **vs. Neural Systematic Binder (NSB):** We distinguished the binding mechanisms. NSB employs a concatenative strategy (routing information into spatial "block-slots"), whereas VRN operates on holographic representations (VSA), where factors are compressed into a single distributed vector via element-wise operations.
* **Adaptive Resonance Theory (ART):** We added a discussion acknowledging the conceptual link to Grossberg's Adaptive Resonance Theory.

### Summary of Feedback and Consensus
We were very encouraged by the reviewers' reception of these revisions. Following our responses and the updates to the manuscript, we reached a positive consensus in the discussion thread:
* **Reviewer SqXF** noted that the notation "reads much more nicely now" and thanked us for the derivations, stating they **had raised their score accordingly (from 2 to 4)**.
* **Reviewer G7xX** stated that our responses addressed their main concerns regarding the quasi-orthogonality approximation and evaluation setup, explicitly confirming they were **willing to increase their rating (originally 6)**.
* **Reviewer qbU6** acknowledged the promise of the method in correlated regimes and stated they **would raise their score (from 4)**.

We hope this summary is helpful as you evaluate our submission.

---

### Meta-Review · Area_Chair_azm6 · 2026-01-03

**Summary:**

**Summary of contribution**  \
The paper tackles the decomposition problem in hyperdimensional computing by proposing a probabilistic solution, offering differentiability, guaranteed convergence and integration to existing probabilistic machine learning models compared to resonator networks.

**Summary of concerns** \
 All reviewers appreciated the clarity of the paper and recognized its conceptual advancement over resonator networks. However, there is unanimous agreement that the experimental evaluation is preliminary and limited, which does not fully do justice to the proposed idea. Additional supporting evidence is necessary to strengthen the paper’s impact and facilitate wider adoption. Specifically:
* The experimental analysis relies predominantly on synthetic datasets. While this choice is valuable for validating the properties of the proposed method, it is not sufficient on its own, particularly given that the contribution is not purely theoretical.
* The only experiment on a more realistic dataset is conducted on colored MNIST, where the proposed solution is integrated into a VAE framework. This evaluation remains largely qualitative and lacks quantitative comparisons against simple baselines. Furthermore, there is no discussion of hyperparameter selection, their practical determination, or their influence on performance. As noted by Reviewer 4NXR, an analysis of identifiability or disentanglement is also missing. In particular, it is unclear whether the latent variables exhibit semantic binding to object attributes.
* The proposed integration with VAEs shares similarities with the Neural Systematic Binder, as highlighted by Reviewer qbU6. This constitutes an important missing baseline that should be included for comparison.
* Finally, the paper does not provide empirical evidence to substantiate the claim of compositionality in the learned representations. Such evidence could be obtained through evaluations on established benchmarks such as dSprites, Shapes3D, or MPI3D, as suggested by Reviewer qbU6.

**Decision** \
The paper presents an interesting and promising approach for decomposing representations into symbolic vector constituents, with clear potential relevance to the representation learning and neuro-symbolic learning communities. However, the work currently appears to be at a preliminary stage. The experimental methodology should be significantly expanded to include missing baselines and more thorough analyses in order to adequately support the paper’s claims and strengthen its overall contribution.

**Reviewer Concerns:**

During the rebuttal, the authors were able to resolve issues about the clarity of the presentation, the soundness of the proposed solution and its computational complexity. However, the concerns about the quality of the experimental methodology remain largely unaddressed.

**Reviewer Scores:**

Reviewers have already raised their score. However, the scoring would have not increased significantly above the acceptance bar.

---

### Decision · Program_Chairs · 2026-01-26

Reject